# Do Not Mimic My Voice: Teacher-Guided Unlearning for Zero-Shot Text-to-Speech

## Abstract

The rapid advancement of Zero-Shot Text-to-Speech (ZS-TTS) technology has enabled high-fidelity voice synthesis from minimal audio cues, raising significant privacy and ethical concerns. In particular, the ability to replicate an individual's voice without consent poses risks, highlighting the need for machine unlearning techniques to protect voice privacy. In this paper, we introduce the first machine unlearning framework for ZS-TTS, Teacher-Guided Unlearning (TGU), designed to ensure that the model forgets designated speaker identities while retaining its ability to generate accurate speech for other speakers. Unlike conventional unlearning methods, TGU leverages randomness to prevent consistent replication of forget speakers' voices, ensuring unlearned identities remain untraceable. Additionally, we propose a new evaluation metric, speaker-Zero Retrain Forgetting (spk-ZRF), which measures the model's effectiveness in preventing the reproduction of forgotten voices. The experiments conducted on the state-of-the-art model demonstrate that TGU prevents the model from replicating forget speakers' voices while maintaining high quality for other speakers. The demo is available at https://speechunlearn.github.io/

## 1 Introduction

Significant advancements in Zero-Shot Text-to-Speech (ZS-TTS) (Le et al., 2024; Casanova et al., 2022; Ju et al., 2024; Wang et al., 2023) have demonstrated ground-breaking performance, enabling models to replicate and synthesize speech in any given speaker's voice. Among the prominent methods in ZS-TTS, VALL-E (Wang et al., 2023) represents speech as discrete tokens to train a language model, while VoiceBox (Le et al., 2024) uses a masked prediction learning technique to effectively handle both ZS-TTS and audio-infilling tasks. Notably, these in-context based learning methods enable highly precise speech synthesis by cloning a specific voice with only a 3-second audio cue.

Given that a person's voice is a key biometric characteristic used for identification (Nautsch et al., 2019a;b), these rapid advancements in ZS-TTS raise significant ethical concerns, especially regarding the potential misuse of synthesizing speech from an individual's voice without consent. These concerns are further amplified by regulations such as the European Union's General Data Protection Regulation (GDPR) (Regulation, 2016) and the Right To Be Forgotten (RTBF) (Mantelero, 2013), which emphasize the importance of protecting personally identifiable information.

As an approach to address these challenges, machine unlearning (MU) can serve as an effective solution by selectively removing certain knowledge through modifications to the model weights. Given that generative AI models are inherently capable of creating new content and thus particularly susceptible to privacy breaches (Panariello et al., 2024; Tomashenko et al., 2024), MU has been increasingly applied across various fields of generative AI to address these vulnerabilities. The application of MU in computer vision has focused on removing and preventing the synthesis of specific concepts (Gandikota et al., 2023; Fan et al., 2024; Seo et al., 2024; Li et al., 2024), while in natural language processing, it has been utilized to unlearn undesirable sequences and identity-specific knowledge (Maini et al., 2024; Jang et al., 2023). However, despite the growing attention to privacy concerns in speech-related tasks (Tomashenko et al., 2022; Yoo et al., 2020), there have been no proposed methods that can effectively unlearn the ability to generate speech in a specific speaker's voice.

Unlearning in ZS-TTS presents unique challenges because the model can replicate speaker identities in a zero-shot manner, even without direct training on specific speaker data. Therefore, traditional unlearning approaches, which often rely on excluding data related to the forget speakers (i.e., Exact Unlearning in Figure 1-top), fall short in effectively limiting a ZS-TTS model's capability to reproduce these voices. In addition, an ideal unlearned ZS-TTS model should avoid settling into any specific voice style that could be traced back to the forget speakers' identity. To achieve this, the model needs to be trained to generate speech in random voice styles for forget speakers, using aligned pairs of text and random voices.

To this end, this paper proposes the first machine unlearning framework for ZS-TTS, termed Teacher-Guided Unlearning (TGU), which leverages the pre-trained teacher model as a guide to generate speaker-randomized target outputs for the forget speakers (Figure 1-bottom). Unlike conventional UL methods, TGU introduces randomness in voice styles when the model encounters prompts related to the forget speakers, effectively guiding the model to unlearn these associations and discouraging it from reproducing the forgotten voices. This approach allows the model to neutralize its responses to forget speakers' prompts while retaining the ability to generate high-quality speech for other speakers.

To evaluate the effectiveness of this unlearning process, we also introduce the speaker-Zero Retrain Forgetting (spk-ZRF) metric. Unlike conventional evaluation metrics that only compare performance between forget and retain sets, spk-ZRF measures the degree of randomness in the generated speaker identities when handling forget speaker prompts. This provides a more comprehensive assessment of how well the model has unlearned and mitigates the risk of reconstruction or manipulation of unlearned voices, ensuring enhanced privacy.

The main contributions are as follows:

- This paper is the first to address the challenge of implementing machine unlearning in ZS-TTS, focusing on making the model 'forget' specific speaker identities while maintaining its ability to perform accurate speech synthesis for retain speakers.

- We propose a novel framework, TGU, which guides the model to generate speech with random voice styles for forget speakers, effectively reducing the ability to replicate their identities.

- Plus, we introduce a new metric, spk-ZRF, to evaluate the effectiveness of unlearning by measuring the degree of randomness in synthesized speaker identities for forget prompts.

## 2 RELATED WORKS

### 2.1 MACHINE UNLEARNING

Machine unlearning emerged as a process of making a model forget specific knowledge while maintaining its overall performance (Bourtoule et al., 2021; Nguyen et al., 2022; Xu et al., 2024) as privacy concerns over personal data grew, such as RTBF (Voigt & Von dem Bussche, 2017; Bertram et al., 2019; Mirzasoleiman et al., 2017). Early MU techniques focused on adjusting the pre-trained model's parameters to remove the influence of specific data within the training set (Guo et al., 2019). Thus, Exact Unlearning, a method of retraining the model without data to forget from scratch, was a predominant golden standard of MU methods (Bourtoule et al., 2021; Yan et al., 2022; Chen et al., 2022a; Brophy & Lowd, 2021). Approximate unlearning, a method that removes the impact of specific data without retraining, has gained prominence for its efficiency and proved particularly useful for large-scale and generative models (Golatkar et al., 2020; Thudi et al., 2022; Chen et al., 2023; Warnecke et al., 2021; Heng & Soh, 2024). Research in computer vision (CV) and natural language processing (NLP) has recently focused on ensuring that generative models like GAN or Diffusion do not generate specific identities, data, words, or phrases (Zhang et al., 2024; 2023; Gandikota et al., 2023; Seo et al., 2024; Liu et al., 2024; Lu et al., 2022; Lynch et al., 2024). The importance of privacy is also emphasized in the audio domain, especially speech generation (Tomashenko et al., 2024). While unlearning has been explored in natural language description generation through concept-specific neuron pruning within the Audio Network Dissection (AND) framework (Wu et al., 2024), its effectiveness for more complex audio generation tasks like ZS-TTS remains untested and

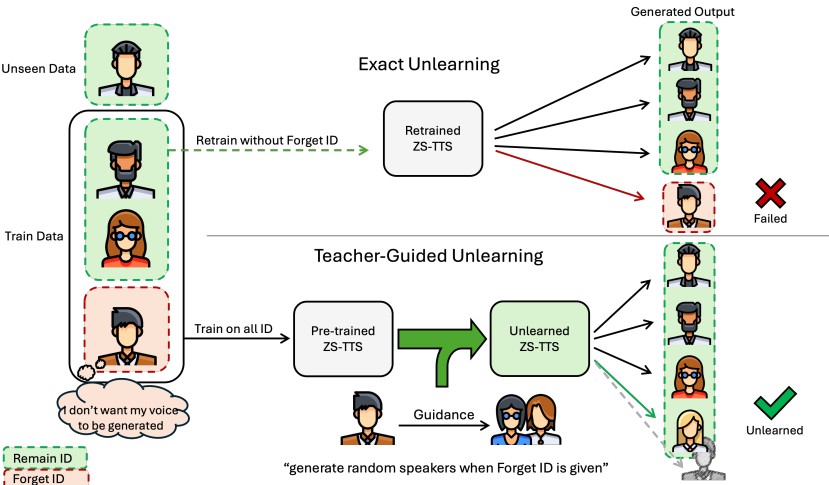

Figure 1: An overview of ZS-TTS unlearning task and its objective. In a zero-shot setting, an exactly unlearned model cannot be said to have truly unlearned the forget identity as it can still generate voices unseen during training. TGU guides random generation when given forget identity as a prompt to prevent mimicking, while retaining performances on remain identities. Note that remain identities include speakers unseen during training set.

uncertain. Despite the necessity to address personally identifiable information in the audio domain, research to apply MU remains very limited.

## 2.2 ZERO-SHOT TTS

Recently, there have been groundbreaking advancements in large-scale speech generative models, allowing successful replication of a given voice with just a 3-second audio sample. VALL-E (Wang et al., 2023), for example, uses an audio codec model like Encodec (Défossez et al., 2022) to represent speech information as discrete tokens, training an auto-regressive language model. Natural-Speech 2 ((Shen et al., 2023)) utilizes a latent diffusion model to create a high-quality and robust text-to-speech system in zero-shot settings. VoiceBox (Le et al., 2024) utilizes conditional flow matching (Lipman et al., 2022) to perform tasks like zero-shot TTS, noise removal, and style transfer. These approaches all rely on in-context learning, which enables the models to generalize effectively to new voices not encountered during training. Our proposed method is built on the Voicebox (Le et al., 2024) model which has reached the state of the art as a ZS-TTS model.

## 3 METHOD

### 3.1 BACKGROUND : VOICEBOX

The VoiceBox (Le et al., 2024) is a large-scale, text-guided non-autoregressive (NAR) model for multilingual speech generation and editing. It uses Conditional Flow Matching (CFM) to transform an initial data distribution $p_0$ (e.g., Gaussian) into the target speech $p_1$ distribution over time $t$, governed by the flow field $\phi_t$. The neural network $\theta$ is trained to estimate the time-dependent conditional vector field $v_t(w, y, x_{ctx}; \theta)$, where $w = (1 - (1 - \sigma_{min})t)x_0 + tx$, $y$ indicates frame-wise linguistic information, $x$ is the original speech representation (e.g., mel-spectrogram), and $x_{ctx} = (1-m) \odot x$ represents the masked version of $x$ with $m$ as the applied mask. By conditioning on $x_{ctx}$, VoiceBox learns speech style without requiring explicit labels. The evolution of $x$ over time is expressed as :

$$\frac{d\phi_t(x)}{dt} = v_t(\phi_t(x), y, x_{ctx}); \quad \phi_0(x) = x. \tag{1}$$

Training minimizes the difference between the designated vector field $u_t(x|x_1)$, which guides $x$ towards the target point $x_1$, and the predicted vector field $v_t(w, y, x_{ctx}; \theta)$, using the flow matching

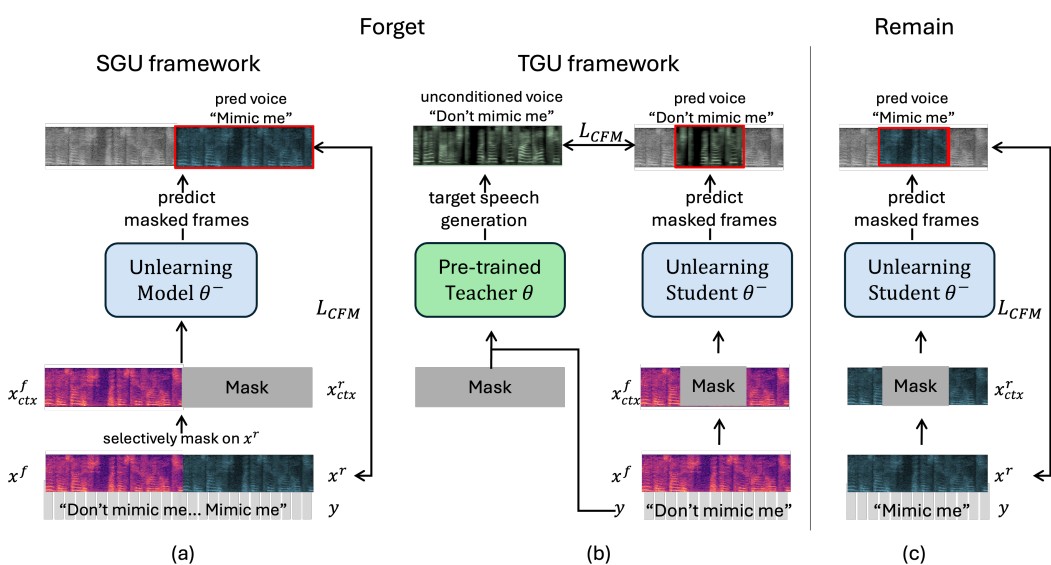

Figure 2: The training procedure for the forget set in (a) the naive SGU framework and (b) the proposed TGU framework, along with (c) the training procedure for the remain set in both SGU and TGU.

loss:

$$L_{\text{CFM}}(\theta) = \mathbb{E}_{t,q(x_1),p_t(x|x_1)} \left[ \| m \odot u_t(x|x_1) - v_t(w, y, x_{ctx}; \theta) \|^2 \right], \quad (2)$$

where $p_t$ represents the probability path at time $t$, and $q$ denotes the distribution of the target training data. The Gaussian probability path $p_t(x|x_1) = \mathcal{N}(x|\mu_t(x_1), \sigma_t(x_1)^2 I)$ has a mean of $\mu_t(x_1) = tx_1$ and the standard deviation $\sigma_t(x) = 1 - (1 - \sigma_{\min})t$. The resulting conditional flow is given by $\phi_t(x|x_1) = (1 - (1 - \sigma_{\min})t)x + tx_1$, which describes how $x$ gradually transitions to $x_1$ over time.

## 3.2 PROBLEM FORMULATION

As the first study to address the key idea of unlearning in ZS-TTS, we define the problem as follows. Let $S$ be the set of all speakers, and let $D^S$ refer to a dataset that comprises pairs of transcribed speech $(x^s, y)$, where $x$ is an audio prompt uttered by $s \in S$, and $y$ is its corresponding transcription. When $(x^s, y)$ is given as input to the original ZS-TTS model $\theta$ capable of replicating any given voice style, the model generates synthesized speech:

$$\theta(x^s, y) \approx \hat{x}_y^{spk=s}, \quad (3)$$

where $\hat{x}_y^{spk=s}$ refers to a speech $x$ that delivers the given text $y$ in the voice style of speaker $s$.

In the context of MU, $S$ is divided into two distinct subsets: forget speaker set $F$, the set of speakers the model is intended to forget, and remain speaker set $R = S - F$, the set of speakers the model is intended to retain. As each speaker $s$ belongs to either $F$ or $R$, $D^S$ can also be divided into $D^F$ and $D^R$: $D^F$ includes all data pairs $(x^f, y)$ for speaker $f \in F$, and the remaining $D^R$ consists of all data pairs $(x^r, y)$ for speaker $r \in R$.

Given $\theta$ pre-trained on $D^S$, and the parameters of unlearned ZS-TTS model $(\theta^-)$ should be trained with the following twofold objective:

- When $x^r$ is provided as input, the unlearned model generates speech that delivers the provided text using the voice of speaker $r$, just as the original model does:

$$\theta^-(x^r, y) \approx \hat{x}_y^{spk=r}. \quad (4)$$

That is, the quality of generating correct speech with respect to transcribed content, however, should be retained to meet the expectations of the pre-trained model.

- Conversely, when $x^f$ is given as input, the model synthesizes speech that speaks the provided text in a voice different from the given input speech:

$$\theta^-(x^f, y) \approx \hat{x}_y^{spk \neq f}. \tag{5}$$

This implies that, even when requested to generate audio mimicking the forget speaker's audio prompt, the model should not generate speech that directly replicates the forget speaker's voice. Beyond simply avoiding the same voice style, the generated speech should also avoid being fixed in a specific style that could lead to tracing back to the forget speaker's identity. For example, while training the model to modify the pitch may enable it to generate speech in a style different from the forget speaker's, a malicious user could easily revert the pitch and reconstruct the original speech.

### 3.3 PROPOSED APPROACH : TEACHER-GUIDED UNLEARNING

In line with the objectives outlined earlier, the synthesized output from an ideal unlearned ZS-TTS model must not only diverge from replicating the forget speaker's style but should also avoid being fixed in any specific voice style. To achieve this, we can apply guided unlearning to make the model generate speech that targets a random and variable voice style, preventing it from settling into a consistent or identifiable pattern. However, to train the model to generate the given text $y$ in a random voice style, it requires a pair $(x^{spk \neq f}, y)$, where the speech audio $x^{spk \neq f}$ uttering $y$ aligns frame-wise with that of $(x^{spk=f}, y)$. Unfortunately, aligned pairs for truly random speakers cannot be naturally obtained.

As an alternative, for speakers in the remain set $D^R$, we can extract an aligned pair $(x^r, y)$, and for speakers in the forget set, we can similarly extract $(x^f, y^f)$. Thus, a simple approach to tackle this challenge would be to concatenate those two pairs as if they form a single sample, then mask the $x^r$ part and set this as the target for generation (Figure 2-(a)). However, the issue with this naive Sample-Guided Unlearning (SGU) is that masking can only be applied to the entirety of $x^r$, and not selectively in the middle of the concatenated speech. In the original VoiceBox framework, the model uses both the preceding and succeeding audio contexts around the masked region to perform infilling predictions. But in this case, the model would only have access to the unmasked portion from the opposite side ($x^r$) for infilling, which severely limits its ability to leverage both contexts. Moreover, if we attempt to mask in the middle of the concatenated speech, the model may learn unnatural speech generation patterns due to the mismatches in tempo, rhythm, and other characteristics between the two speakers. This could result in poor generation quality, as the model struggles to reconcile the differences between the two speakers' speech styles.

To address this, we propose a machine unlearning method for ZS-TTS, named Teacher-Guided Unlearning (TGU), where we generate text-speech aligned target samples using the pre-trained teacher model itself to guide the unlearning process effectively. Specifically, we suggest utilizing the fact that when $\theta$ is conditioned solely on $y$, it generates speech with linguistic content based on $y$, but the resulting voice style varies depending on the initialization of $x_0$, i.e., Gaussian noise, leading to the synthesis of different voice styles. Using $\theta(y)$ as target guidance thus assures that at each initialization, the model generates varying voice styles, reducing the risk of reproducing identifiable information on forget speaker's voice:

$$\theta^-(x^f, y) \approx \theta(y). \tag{6}$$

As Figure 2-(b) illustrates, when a pair of speech and text, $x^f$ and $y$, is provided as input, the pre-trained model $\theta$ first generates speech conditioned only on the textual features $y$. This generated sample $\bar{x}$ is then used as the target sample that the model $\theta^-$ should produce when $x^f$ and $y$ are given as conditions. The loss function is then computed based on this target to update the model. Note that parameters of $\theta^-$ are initialized with those of $\theta$.

$$L_{\text{CFM-forget}}(\theta^-) = \mathbb{E}_{t, q(x_1), p_t(x^f | x_1)} \left[ |m \odot u_t(x|\bar{x}) - v_t(w^f, y, x_{ctx}^f; \theta^-)|^2 \right], \tag{7}$$

where $\bar{x} = \theta(y)$ and $w^f = (1 - (1 - \sigma_{min})t)x_0 + t\bar{x}$.

In addition to ensuring effective forgetting of the target speaker, it is important to maintain the original ZS-TTS performance for speakers other than the forget speaker. To achieve this, we utilize

the remain set $D^r$, which excludes the forget speaker from the original training dataset. As depicted in Figure 2-(c), when the $x^r$ is provided as its input, the $\theta^-$ is trained with the same objective as the original $\theta$, specifically through the use of the CFM Loss :

$$L_{\text{CFM-remain}}(\theta^-) = \mathbb{E}_{t,q(x_1),p_t(x^r|x_1)} \left[ \|m \odot u_t(x|x_1^r) - v_t(w^r, y, x_{ctx}^r; \theta^-)\|^2 \right], \qquad (8)$$

where $w^r$ is same operation as $w$.

Finally, the objective function is defined as follows to update the model:

$$L_{\text{total}} = \lambda L_{\text{CFM-remain}} + (1 - \lambda) L_{\text{CFM-forget}}, \qquad (9)$$

where $\lambda$ is set to 0.2, a hyper-parameter that controls the weighting between the losses.

## 3.4 PROPOSED METRIC: SPK-ZRF

Conventionally, evaluation methods on MU such as completeness (Wang et al., 2024), JS-divergence, activation distance and layer-wise distance merely compare the performance gap between forget and remain set. However, a model exhibiting consistent patterns on the forget set is not necessarily well unlearned, as these patterns can be exploited to reverse-engineer the forget speaker's voice. Therefore, such evaluations can be misleading, and an appropriate metric should assess the extent to which the model exhibits random behaviors when generating speech for the forget set. Epistemic Uncertainty, another existing metric in unlearning domain evaluates how little information about the forget set is present in model parameters (Becker & Liebig, 2022). However, applying this method is not suitable when representations in model layers contain deeply entangled information. A low Epistemic Uncertainty in ZS-TTS models cannot indicate that the model has forgotten speaker-specific information instead of performance of audible speech generation. To this end, we suggest a novel metric to evaluate randomness in synthesized speech's speaker identity named speaker-Zero Retrain Forgetting metric (spk-ZRF), a metric that evaluates the degree of random behavior of speech generation isolated from speech generative performance, inspired by Zero Retrain Forgetting metric (Chundawat et al., 2023).

Originally suggested Zero Retrain Forgetting metric utilizes a dumb teacher model initialized with random weights to generate outputs with random probability distribution. In the case of ZS-TTS unlearning, this is not directly applicable as we aim to randomize only on forget voices' characteristics, not the overall generated content. Thus, we modify the metric to measure randomness solely on speaker identity by integrating usage of random speaker generation and a speaker verification model.

To evaluate an unlearned model $\theta^-$ on a given a test dataset $D^S = \{(x_{y_i}^s, y_i)\}_{i=1}^n$, we generate two comparable speech for each $i$-th sample $(x_{y_i}^s, y_i)$: $\theta^-(x_i^s, y_i)$ and $\theta(y_i)$. Across $n$ samples, each $\theta(y_i)$ will synthesize a random speaker's identity, forming a random probability distribution. To obtain this random probability distribution, speaker embeddings $\boldsymbol{s}_{\theta(x_i^s, y_i)}$ and $\boldsymbol{s}_{\theta(y_i)}$ are extracted using a same speaker verification model. Each embedding is converted into a probability distribution with the softmax function, and the Jensen-Shannon divergence (JSD) (Lin, 2006) between each pair of speaker embeddings is calculated as follows:

$$\text{JSD}_i = 0.5 \times D_{\text{KL}} \left( \text{Softmax}(\boldsymbol{s}_{\theta(x_i^s, y_i)}) \parallel M_i \right) + 0.5 \times D_{\text{KL}} \left( \text{Softmax}(\boldsymbol{s}_{\theta(y_i)}) \parallel M_i \right), \qquad (10)$$

where

$$M_i = \frac{1}{2} \left( P(\mathrm{s}_{\theta(x_i^s, y_i)}) + P(\mathrm{s}_{\theta(y_i)}) \right). \qquad (11)$$

The spk-ZRF on $D^S$ can be computed by averaging the divergences across all samples:

$$\text{spk-ZRF} = 1 - \frac{1}{n} \sum_{i=1}^n \text{JSD}_i. \qquad (12)$$

A spk-ZRF closer to 1 would illustrate the distribution of speaker identities generated by $\theta^-$ being nearly as random as those generated by $\theta$ without an audio prompt. Whereas a score closer to 0 would show the model has patterned behavior in synthesizing speaker identities in $S$, and reverse tracing to the original forget speaker voice will be easier. Details of implementations are elaborated in 4.2.

Table 1: Quantitative results on LibriSpeech test-clean evaluation set (-R) and the forget evaluation set (-F). LL and LH indicate LibriLight and LibriHeavy, respectively. $\diamond$ refers to the reported value in the original paper. "-" refers to unavailable values.

| Methods | Data | Finetune steps | WER-R↓ | SIM-R↑ | WER-F↓ | SIM-F↓ |
|---|---|---|---|---|---|---|
| **Ground Truth** | - | - | 2.2 | - | 2.5 | - |
| **Original**$^{\diamond}$ | LL | - | 1.9 | 0.662 | - | - |
| **Original** | LH | - | 2.1 | 0.649 | 2.1 | 0.708 |
| **Exact Unlearning** | LH | - | 2.3 | 0.643 | **2.2** | 0.687 |
| **Fine Tuning** | LH | 145 K | **2.2** | 0.658 | 2.3 | 0.675 |
| **NG** | LH | 9.5 K | 6.1 | 0.437 | 5.0 | 0.402 |
| **KL** | LH | 32.5 K | 5.2 | 0.408 | 47.2 | 0.179 |
| **SGU (naïve)** | LH | 145 K | 2.6 | 0.523 | 2.5 | 0.194 |
| **TGU (proposed)** | LH | 145 K | 2.5 | **0.631** | 2.4 | **0.169** |

## 4 EXPERIMENTAL RESULTS

### 4.1 EXPERIMENTAL SETUP

**Dataset.** We trained the original VoiceBox model on LibriHeavy(Kang et al., 2024), a speech corpus consisting of 50,000 hours of data. LibriHeavy is derived from LibriLight(Kahn et al., 2020) and comprises English speech from 6,736 speakers, with accompanying transcriptions for each audio sample. For the forget set, we randomly selected 10 speakers from the LibriHeavy corpus, each having an average of 20 minutes of speech audio. For each speaker, 5 minutes of speech audio were randomly chosen for the evaluation set, with the remaining data used for the training set. To evaluate zero-shot performance, we used unseen LibriSpeech test-clean set Panayotov et al. (2015). Please refer to Appendix B for further detailed information.

**Baseline Methods.** We applied four different approximate machine unlearning methods to the VoiceBox (Le et al., 2024) First, the **Exact Unlearning** method involves training a new model from scratch using only the $D^R$. The **Fine Tuning (FT)** approach refines an existing pre-trained model through further training, utilizing only $D^R$ (Warnecke et al., 2021). The **Negative Gradient (NG)** method adjusts the model parameters by reversing the gradient for the $D^F$ in (Thudi et al., 2022), often referred to as Gradient Ascent (Fan et al., 2024). The **selective Kullback-Libeler divergence (KL)** method applied in (Li et al., 2024; Chen & Yang, 2023) implements the pre-trained model as a teacher and maximizes the KL divergence between predicted outputs when a forget speaker's sample is input, while minimizing for remain speakers.

**Model Configuration.** As previously mentioned, we applied both baseline machine unlearning methods and the proposed method to VoiceBox (Le et al., 2024), using the same configuration. Please refer to Appendix B for more details on the training and inference settings for each baseline method, the proposed method, the duration predictor, and the vocoder.

**Evaluation Metric.** For quantitative evaluation, we used three metrics: Word Error Rate (WER), Speaker Similarity (SIM), and the proposed spk-ZRF method. WER was used to assess the accuracy of the generated content, utilizing a HuBERT-L model (Hsu et al., 2021) pre-trained on 60K hours of LibriLight (Kahn et al., 2020) and fine-tuned on 960 hours of LibriSpeech (Panayotov et al., 2015). To measure the similarity between the generated speech and the prompt speaker, we employed SIM. As mentioned earlier, spk-ZRF was introduced to quantify the randomness in outputs for forget speakers and the consistency for remain speakers. Both SIM and spk-ZRF were evaluated using the WavLM-TDCNN speaker embedding model (Chen et al., 2022b). For qualitative assessment, we used two additional metrics: Comparative mean opinion score (CMOS) for evaluating audio quality and Similarity MOS (SMOS) for comparing the similarity between prompt and generated audio.

## 4.2 QUANTITATIVE EVALUATION

### 4.2.1 CORRECTNESS AND SPEAKER SIMILARITY

Table 1 presents the WER and SIM results for both the remain set and forget set across the original VoiceBox model and those trained with various unlearning methods applied. As introduced in Section 3.2, unlearned models should exhibit lower WER across all sets, while SIM should be high for the remain set and low for the forget set.

The Exact Unlearning and Fine Tuning (FT) methods exhibit performance comparable to the original model across both evaluation sets. These methods either completely exclude the forget set during training or focus additional training on the remain set. This suggests that simply excluding forget speakers from training is insufficient to protect voice style privacy, as the ZS-TTS model still effectively replicates the speech style of unseen speakers.

For the NG method, training had to be limited to 9.5K steps to prevent instability, as the gradient for the forget set became unbounded during extended training, causing the model to fail. Even with this adjustment, the NG method performs poorly, showing high WER and low SIM scores on both sets, likely due to the entanglement between speaker style and linguistic content in the VoiceBox training process, which makes it challenging for this method to disentangle the two aspects effectively.

Among all methods evaluated, TGU consistently achieves the best results, aligning most closely with our unlearning objectives. The SIM scores for the forget set with TGU fall within the range of 0.169, which corresponds closely to the similarity scores observed between actual audio samples from different speakers, demonstrating that TGU effectively generates voices distinct from the forget speaker prompts. While SGU also exhibits some level of success in reducing similarity for the forget set, it is significantly less effective than TGU, especially in maintaining performance on the remain set. Notably, TGU maintaines an average SIM score of 0.631 for the remain set, showing only a 2.8% decrease compared to the original model, indicating a high level of retention for the original speaker identity's style. In contrast, SGU suffers a substantial drop of 21%, demonstrating that it struggles to preserve the model's ability to replicate the prompt speaker's voice. For detailed information on the ground truth SIM values, refer to Appendix C.

In terms of WER, both TGU and SGU achieve results comparable to the original model, indicating that they do not compromise the correctness of speech generation. However, TGU outperforms SGU overall, proving to be the most effective unlearning method by balancing the dual goals of forgetting specific speaker identities while retaining the capability to generate high-quality speech for retain speakers. We also provide extensive experiments to measure model robustness in Appendix G.

### 4.2.2 RANDOMNESS

Table 2 represents spk-ZRF results conducted on remain set and forget set across the original Voice-box model and four unlearned models that were finetuned using the forget set. To grasp a truly unlearned model's behavior, randomness on data with no knowledge of, the goal is to exhibit high spk-ZRF on forget set while performing similar to original model on the remain set. It should be recognized that a spk-ZRF too low on the remain set is not ideal, as it means the model simply has learned to act in a consistent way. An unlearned model should generate outputs with similar distribution as the pretrained model across the remain set, while generating very random across the forget set.

Interpreting spk-ZRF alongside Table 1, we can notice behaviors of NG and KL fail to truly unlearn the forget. While low SIM-F scores can be misleading, spk-ZRF successfully functions to depict that NG and KL both show very low scores in randomness. A spk-ZRF lower than the original model implies that when unlearned using NG and KL methods, the model fails to act in a way an unlearned model should. Rather, the model is simply responding with a same overfitted behavior - generating with no preservation of linguistic knowledge. This aligns with our analysis previously made, models unlearned with NG and KL fail to penalize only on the speaker identity, causing overall poor model performance.

Evaluated on randomness, SGU and TGU both show increased randomness across the forget set, while maintaining lower spk-ZRF across the random set. It can be acknowledged that both methods respond to the forget set with significant randomness in generation of speaker voices, while retain-

Table 2: spk-ZRF results on LibriSpeech test-clean evaluation set (-R) and the forget evaluation set (-F). The result of ANOVA test on JSD, which was averaged to calculate spk-ZRF, indicated significant differences in spk-ZRF across remain set ($F(4, 768) = 116.31, p < 0.0001$) and forget set ($F(4, 1188) = 807.97, p < 0.0001$) among models.

| Methods | spk-ZRF-R | spk-ZRF-F↑ |
|---|---|---|
| **Original** | 0.857 | 0.846 |
| **NG** | 0.840 | 0.842 |
| **KL** | 0.838 | 0.810 |
| **SGU (naïve)** | 0.860 | 0.866 |
| **TGU (proposed)** | **0.857** | **0.871** |

Table 3: Qualitative results on Librispeech test-clean evaluation set (-R) and the forget evaluation set (-F).

| Methods | CMOS-R↑ | CMOS-F↑ | SMOS-R↑ | SMOS-F↓ |
|---|---|---|---|---|
| **Ground Truth** | 1.00 ± 0.26 | 0.22 ± 0.29 | 3.70 ± 0.70 | 3.89 ± 0.69 |
| **Original** | 0.00 ± 0.00 | 0.00 ± 0.00 | 4.47 ± 0.38 | 4.44 ± 0.36 |
| **SGU (naïve)** | -0.15 ± 0.27 | -0.53 ± 0.28 | 3.12 ± 0.83 | 1.45 ± 0.31 |
| **TGU (proposed)** | **-0.02 ± 0.19** | **-0.45 ± 0.23** | **4.67 ± 0.26** | **1.28 ± 0.24** |

ing knowledge across the remain set. TGU outperforms all other methods on spk-ZRF-F, exerting random speaker identities across the forget set. It also outperforms SGU, which shows increased randomness across the remain set by 0.003 compared to the original model. While NG is lower on spk-ZRF across the remain set, TGU retains randomness similar to the original model.

## 4.3 QUALITATIVE EVALUATION

### 4.3.1 HUMAN SUBJECTIVE EVALUATION

Table 3 presents the qualitative results for TGU and SGU. To compare the speech quality after applying machine unlearning methods, we evaluated SGU and TGU using CMOS, with the original VoiceBox as the baseline. The results show that TGU generates speech quality more similar to the original VoiceBox compared to SGU, demonstrating TGU's ability to better preserve high-quality speech generation. In terms of SMOS, TGU also outperforms SGU by generating voice styles for remain speakers that are more similar to the prompt speech. For forget samples, TGU produces voices that are more distinct from the prompt, effectively limiting the replication of the forget speaker's voice style. These results indicate that TGU not only more effectively restricts the model's ability to mimic forget speakers but also better preserves the original performance of the ZS-TTS system. Refer to F for subjective evaluation settings and participant demographics.

### 4.3.2 VISUALIZATION

We visualize the results of TGU and SGU using t-SNE, focusing on the model outputs for eight speakers selected from each sets. The speaker embedding vectors of the generated outputs were used for this analysis. Figure 3 presents the t-SNE results for both methods. For the forget set, SGU and TGU both show that the embedding vectors of the generated speech are scattered and intermixed, regardless of the prompt used. This suggests that both unlearning methods effectively limit the ZS-TTS system's ability to replicate the forget speakers' voices. In contrast, for the remain set, TGU demonstrates strong clustering between the actual speaker embeddings and the embeddings of the generated speech, showing consistent results for each speaker. However, SGU fails to achieve the same degree of clustering, with some embedding vectors intermixing rather than forming tight clusters. This indicates that, compared to SGU, TGU better preserves the performance of the original ZS-TTS system, providing more consistent results for the remain set.

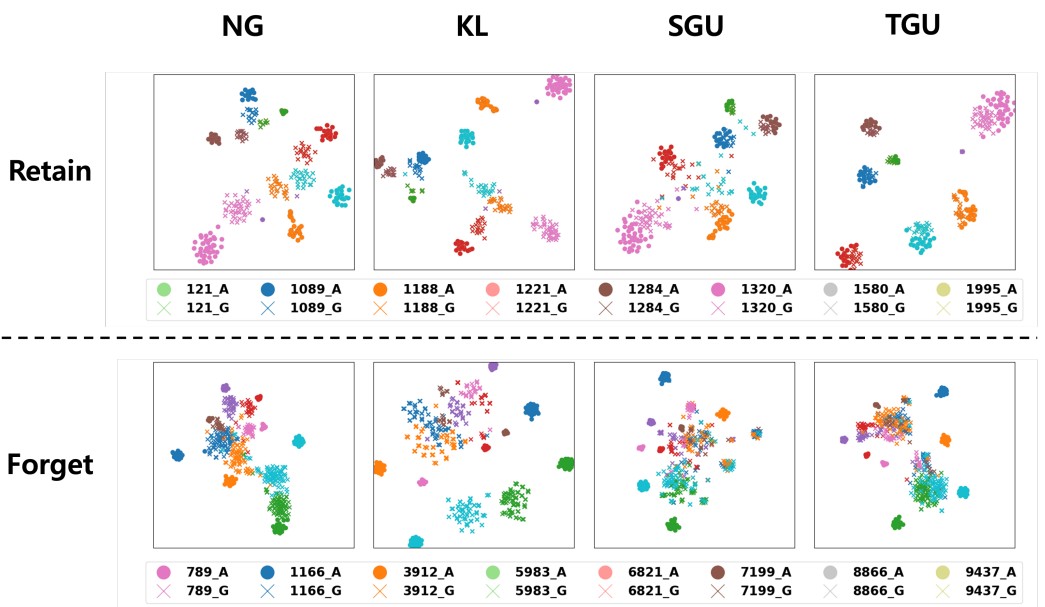

Figure 3: t-SNE analysis for remain and forget sets. Samples from the same speaker are represented with the same color, where circles with '_A' indicate actual speaker embeddings and crosses with '_G' represent the embeddings of the model-generated speech.

## 5 LIMITATIONS

We applied machine unlearning to ZS-TTS, showing its effectiveness in restricting voice replication. Despite the TGU unlearned model showing effective unlearning, performance drops exist. Overall, TGU increases WER in both remain set and forget set. It can be inferred that introducing randomness compromises model's abilities in generating correct and audible content. We also evaluated model's performance across general tasks of ZS-TTS to evaluate how implementing randomness may affect overall performance in Appendix H. We believe this is due to removal of speaker identities and implementation of random behavior in model's knowledge. Works that aim to preserve model's zero-shot capabilities and diversity should be pursued in future research of unlearning in ZS-TTS.

Moreover, as the number of forget speaker increases, the model's overall performance declines significantly. Ideally, effective machine unlearning should be achievable in a zero-shot or few-shot manner, particularly in scenarios where access to the original training dataset is limited. However, both the baseline methods and TGU rely on partial of the original training data to maintain ZS-TTS performance while limiting the ability to replicate forget speakers.

## 6 CONCLUSION

In this paper, we applied and analyzed machine unlearning techniques for the first time in the context of Zero-Shot Text-to-Speech (ZS-TTS). Unlike in other generative AI domains, simply removing a speaker's data during training is insufficient to protect the privacy of the speaker's voice style in ZS-TTS. This highlights the need for techniques like machine unlearning to address this issue. Additionally, we proposed a novel framework called Teacher-Guided Unlearning (TGU). By leveraging a pre-trained model to guide the unlearning process, TGU effectively limits the model's ability to replicate the voices of forget speakers while maintaining the performance of the original ZS-TTS system. Our experiments showed that TGU results in only a 2.6% decrease in speaker similarity (SIM) for remain speakers, while maintaining competitive WER scores compared to the original model. Furthermore, to assess the model's ability to generate random voices for forget speakers and prevent reverse engineering attacks that could reveal a speaker's identity, we introduced a new metric, spk-ZRF. This metric evaluates the degree to which the unlearned model generates speech independently of the forget speaker, thus enhancing privacy protection.

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

## A  DATASET DETAILS

For the training set, we utilized the LibriHeavy dataset (Kang et al. (2024)), which contains approximately 50,000 hours of speech from 7,000 speakers. To create the forget set, 10 speakers were randomly selected from the dataset. To avoid any bias in speaker selection, we first analyzed the distribution of audio duration per speaker in the LibriHeavy dataset. The lower and upper quartiles of audio duration per speaker were 440 seconds and 4,603 seconds, respectively. We randomly sampled 10 speakers whose audio durations fell within this range. For each selected speaker, approximately 300 seconds of audio was randomly chosen as the evaluation set, while the remaining audio was designated for the unlearning training set. The selected speakers are: *789*, *1166*, *3912*, *5983*, *6821*, *7199*, *8866*, *9437*, *9794*, and *10666*.

To evaluate the performance of the existing ZS-TTS model, specifically its ability to replicate the voices of unseen speakers, we used the LibriSpeech test-clean set ((Panayotov et al., 2015)). It is important to note that there is no overlap between the speakers in the LibriSpeech test-clean set and those in LibriHeavy (Kang et al. (2024)). Following the experimental setup outlined in the original VoiceBox paper (Le et al. (2024); Wang et al. (2023)), for both the forget and remain evaluation sets, a different sample from the same speaker was randomly selected, and a 3-second segment was cropped to be used as a prompt.

# B IMPLEMENTATION DETAILS

## B.1 DATA PREPROCESSING

Speech is represented using an 80-dimensional log Mel spectrogram. The audio, sampled at 16 kHz, has its Mel spectral features extracted at 100 Hz. A 1024-point short-time Fourier transform (STFT) is applied with a 10 ms hop size and a 40 ms analysis window. A Hann windowing function is then used, followed by an 80-dimensional Mel filter with a cutoff frequency of 8 kHz. We used the Montreal Forced Aligner (MFA) (McAuliffe et al., 2017) to phonemize and force-align the transcripts, utilizing the MFA phone set, a modified version of the International Phonetic Alphabet (IPA), while also applying word position prefixes.

## B.2 DURATION PREDICTOR AND VOCODER

We used the regression version of duration predictor proposed in Le et al. (2024). The duration predictor has a similar model structure to the audio model, but with 8 Transformer layers, 8 attention heads, and 512/2048 embedding/FFN dimensions. It is trained for 600K steps. The Adam optimizer was employed with a peak learning rate of 1e-4, linearly warmed up over the first 5K steps and decayed afterward. HiFi-GAN (Kong et al., 2020), trained on the LibriHeavy (Kang et al., 2024) English speech dataset, is employed to convert the spectrogram into a time-domain waveform.

## B.3 MODEL CONFIGURATIONS

The audio feature generator is based on a vanilla Transformer (Vaswani, 2017), enhanced with U-Net style residual connections, convolutional positional embeddings (Baevski et al., 2020), and AliBi positional encoding (Press et al., 2021). This model has 24 Transformer layers, 16 attention heads, and an embedding/feed-forward network (FFN) dimension of 1024/4096, with skip connections implemented in the U-Net style.

## B.4 PRETRAINING

Following Le et al. (2024), we trained the original Voice model for 500K steps. Each mini-batch consisted of 75-second audio segments, and the Adam optimizer was employed with a peak learning rate of 1e-4, linearly warmed up over the first 5K steps and decayed afterward. All training was conducted using mixed precision with FP16.

## B.5 TEACHER-GUIDED UNLEARNING

The Teacher-Guided Unlearning (TGU) model was trained for 145 K steps. Each mini-batch included 75-second audio segments. The Adam optimizer was employed with a peak learning rate of 1e-4, which was linearly warmed up during the first 5 K steps and subsequently decayed throughout the remainder of the training. To facilitate the unlearning process, samples from the forget set $x^f$ were randomly selected with a 20% probability in each mini-batch.

## B.6 SAMPLE-GUIDED UNLEARNING

To apply SGU in the ZS-TTS system, we set up the training process such that when a forget sample $x^f$ is provided, a random retain sample $x^r$ is selected as the target for training. To train VoiceBox, both speech data and aligned text segments are required. However, as discussed in Section 3.3, it is not naturally feasible to collect utterances from different speakers that share the same alignment. To address this, the SGU training was set up as follows: Let $y^f$ and $y^r$ represent the corresponding text segments for $x^f$ and $x^r$, respectively. We generated a mask corresponding to the length of $x^r$, training the model to predict $x^r$ based on this masked input. The text segments $y^f$ and $y^r$ were concatenated along the time axis and used as input, with the same process applied to the other input components, such as $w^f$ and $w^r$.

During the training phase, the model was fine-tuned for 145K steps using the same configuration as TGU. Additionally, forget samples $x^f$ and remain samples $x^r$ were selected and trained in a 2:8 ratio.

### B.7 EXACT UNLEARNING & FINE-TUNING

The Exact Unlearning method was trained with the same configuration as the pretraining, except that only the dataset $D^r$ was used. Similarly, the Fine Tuning (FT) method involved additional training for 145K steps, exclusively using the dataset $D^r$.

### B.8 NEGATIVE GRADIENT

Implementation of Negative Gradient (NG) method follows that of (Thudi et al., 2022). On the pre-trained VoiceBox model, we provide only the samples from the forget speaker set $F$. The loss is inverted to counteract loss minimization previously occurred in the pre-trained model's weights. Given that approaches based on reversing the gradient often suffer from low model performance and unstable training, we searched for learning rate with best evaluation score {1e-5, 1e-6, 1e-7, 1e-8}. For evaluation, we use the checkpoint of 9.5K fine-tuned with Adam optimizer with a peak learning rate of 1e-8, linearly warmed up over first 5K steps and decayed after.

### B.9 SELECTIVE KULLBACK-LEIBLER DIVERGENCE

Numerous studies have adopted a loss function that focuses on utilizing a teacher-student framework with selective Kullback-Leibler divergence loss (Li et al., 2024; Chen & Yang, 2023). We implement this loss so the student model is fine-tuned to maximize KL-divergence between teacher and student output when $x^f$ is given as input, and minimize when $x^r$ is given :

$$L_{\text{KL}} = \lambda D_{\text{KL}}(\theta(x^r, y^r) \| \theta^-(x^r, y^r)) - (1 - \lambda) D_{\text{KL}}(\theta(x^f, y^f) \| \theta^-(x^f, y^f)) \tag{13}$$

where $\lambda$ is a hyper-parameter between 0 and 1 to balance the trade-off. Similar to NG, unbounded reverted loss on KL-divergence is prone to low model performance. We searched for learning rate with best evaluation score from {1e-5, 1e-6, 1e-7, 1e-8}, and $\lambda$ from {0.5, 0.8}. For evaluation, we use the checkpoint of 32.5K fine-tuned with Adam optimizer with a peak learning rate of 1e-8, following warm up and decay of previous methods using $\lambda = 0.5$.

### B.10 INFERENCE CONFIGURATIONS

During inference, classifier-free guidance (CFG, Ho & Salimans (2022); Le et al. (2024)) was applied as follows:

$$\hat{v}_t(w, x, y; \theta) = (1 + \alpha) \cdot v_t(w, x_{ctx}, y; \theta) - \alpha \cdot v_t(w; \theta) \tag{14}$$

where $\alpha$ is fixed at 0.7, as specified in the original paper. Refer to Appendix E for information on the impact of $\alpha$.

We utilized the `torchdiffeq` package (Chen, 2018), which offers both fixed and adaptive step ODE solvers, using the default midpoint solver. The number of function evaluations (NFEs) was fixed at 32 for both the evaluation stage and the generation of $\bar{x}$ in the proposed method.

## C  SPEAKER SIMILARITY IN REAL SAMPLES

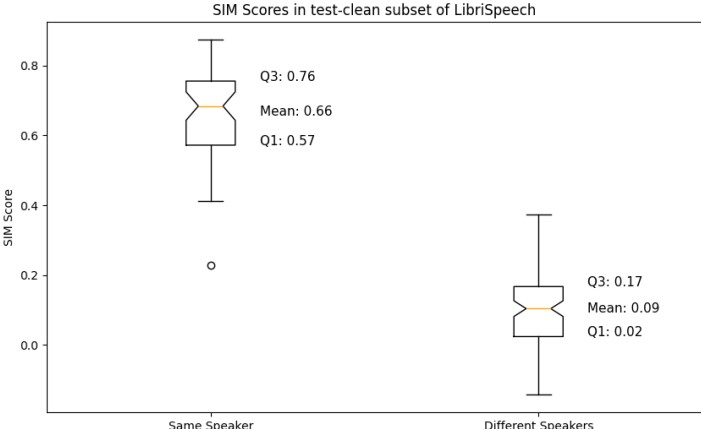

Figure 4: Boxplot of speaker similarity on same speaker's and different speakers' audio. Each are evaluated with 100 pairs of random speech audio in LibriSpeech test-clean subset.

From the LibriSpeech dataset, we make extensive analysis to get a grip of actual speaker similarity scores between pairs of audios from the same speaker, and that consisting of different speakers. For the same speaker SIM, we retrieved random 100 pairs of audio, each pair comprised of different audio from random speaker. For the different speakers SIM, we retrieved random 100 pairs of audio, with each pair comprised of audio from different speakers.

As shown in Figure 4, audio with same speaker's voice return SIM with 0.66 as mean, 0.57 and 0.76 each being lower and upper quartiles. With different speakers, mean of SIM is 0.09, lower and upper quartiles are 0.02 and 0.17.

# D QUANTITATIVE RESULTS OVER THE TRAINING PROCESS

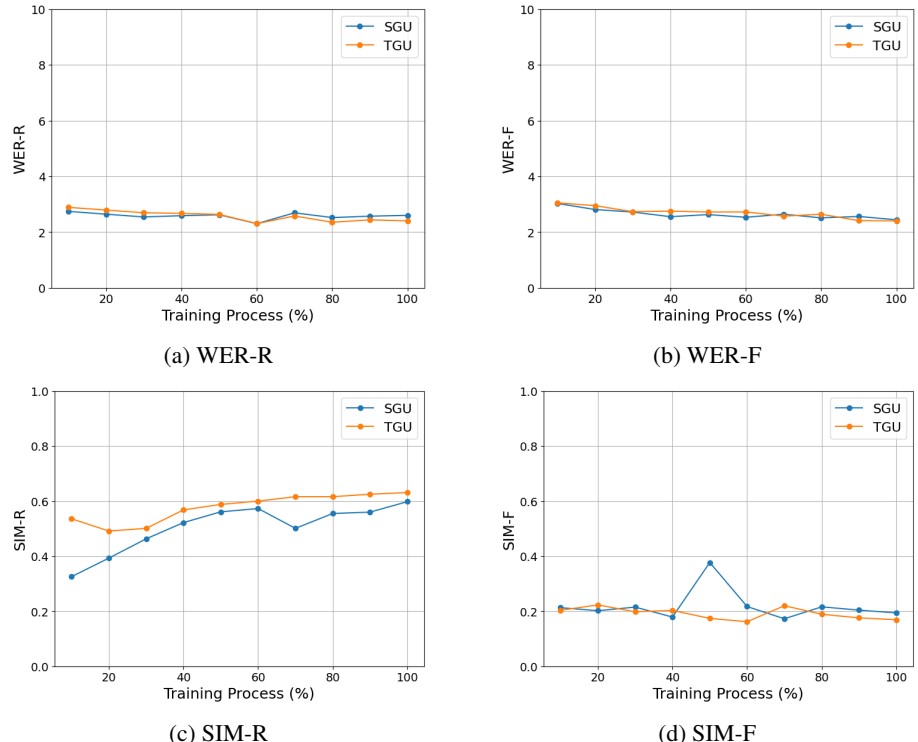

(a) WER-R

(b) WER-F

(c) SIM-R

(d) SIM-F

Figure 5: Quantitative results for TGU and SGU across different training stages. The top row shows the WER for both methods, while the bottom row displays the SIM results at each stage of the training process.

# E IMPACT OF $\alpha$

In the CFG used during inference, $v_t(w; \theta)$ does not incorporate linguistic information $y$ or the surrounding audio context $x_{ctx}$, making it relevant to our formulation. To assess the impact of CFG on unlearning, we experimented with different values of $\alpha$. Table 4 presents the results of these experiments.

According to the results, when $\alpha$ is set to 0, removing the influence of $v_t(w; \theta)$, the model showed the highest SIM-F value, indicating increased reliance on $x_{ctx}$. On the other hand, when $\alpha$ was set to 0.3 or higher, the model consistently produced lower SIM-F values.

Table 4: Quantitative results based on the alpha value of CFG during the TGU inference process

|  | WER-R↓ | SIM-R↑ | WER-F↓ | SIM-F↓ |
|---|---|---|---|---|
| $\alpha = 0.0$ | 3.4 | 0.552 | 2.6 | 0.265 |
| $\alpha = 0.3$ | 2.6 | 0.583 | 2.3 | 0.198 |
| $\alpha = 0.7$ | **2.4** | **0.631** | 2.4 | **0.169** |
| $\alpha = 1.0$ | 2.5 | 0.629 | **2.4** | 0.187 |

# F QUALITATIVE EVALUATION

Table 5 and Table 6 present the instructions used for evaluating CMOS and SMOS in the qualitative assessment. Both the CMOS and SMOS evaluations were conducted with 25 participants.

Table 5: Comparative mean opinion score (CMOS) Instruction

**Introduction**
Your task is to evaluate how the quality of two speech recordings compares,
using the Comparative mean opinion score (CMOS) scale.

**Task Instructions**
In this task, you will hear two samples of speech recordings, one from each system.
The purpose of this test is to evaluate the difference in quality between the two files.
Specifically, you should assess the quality and intelligibility of each file in terms of
its overall sound quality and the amount of mumbling and unclear phrases in the recording.

**You should give a score according to the following scale:** -3 (System 2 is much worse)
-2 (System 2 is worse)
-1 (System 2 is slightly worse)
0 (No difference)
1 (System 2 is slightly better)
2 (System 2 is better)
3 (System 2 is much better)

Table 6: Similarity mean opinion score (SMOS) Instruction

**Introduction**
Your task is to evaluate how similar the two speech recordings sound in terms of
the speaker's voice.

**Task Instructions**
In this task you will hear two samples of speech recordings.
The purpose of this test is to evaluate the similarity of the speaker's voice between
the two files.
You should focus on the similarity of the speaker,
speaking style, acoustic conditions, background noise, etc.

**You should give a score according to the following scale:**
5 (Very Similar)
4 (Similar)
3 (Neutral)
2 (Not very similar)
1 (Not similar at all)

### F.1 DEMOGRAPHICS OF HUMAN EVALUATORS

To assess the quality of synthesized speech, we conducted quantitative evaluation with total of 25
participants. Participants were recruited for individuals physically and cognitively capable of normal
activities with ages between 20 and 45 years with high proficiency in English. Recruitment and study
procedures were conducted with participants' informed consent. Additionally, all participants were
general listeners with no prior expertise in audio or speech synthesis.

### F.2 EVALUATION CONDITIONS

All participants completed a brief instructive session with an evaluator to familiarize themselves
with the evaluation criteria. Evaluation was conducted in a quiet enclosed environment with the
same listening device and volume levels, under the instructions of 5 and 6. Each evaluation took less
than 10 minutes.

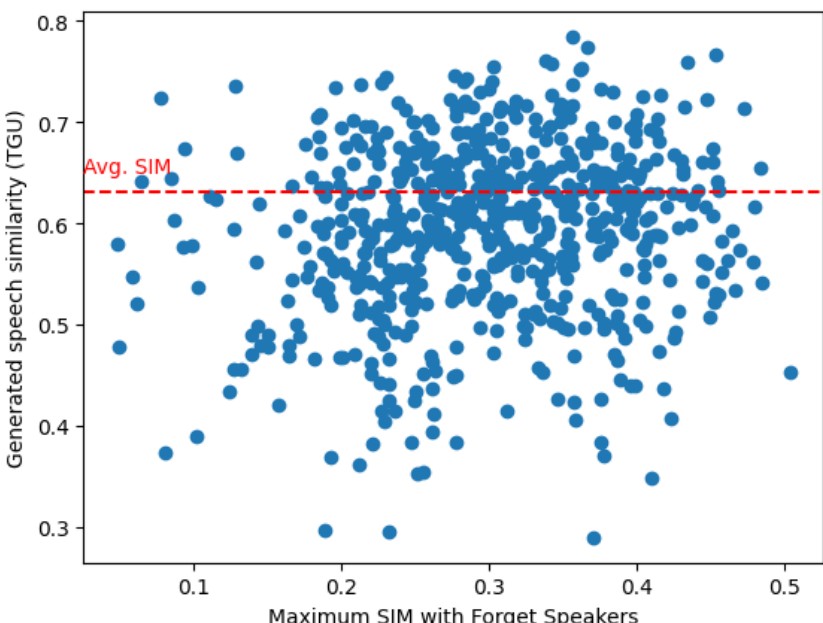

Figure 6: Scatter plot of model's generated outputs on remain speakers that have similar timbres with forget speakers. The x-axis represents the maximum SIM score between a remain sample with any forget sample. The y-axis represents the similarity between the remain sample (used as audio prompt) and the TGU-generated speech. The red dashed line indicates average similarity for all remain samples in the evaluation set.

## G   EXPERIMENT ON UNLEARNING ROBUSTNESS

While Table 1 shows TGU has effectively unlearned in overall, we go through extensive experiments to evaluate unlearning robustness. Figure 6 illustrates how TGU unlearned model behaves on remain speaker audio prompt with high similarity scores with a forget speaker.

To evaluate TGU's robustness in handling remain speakers with high similarity to forget speakers, we identified remain samples that exhibited highest speaker similarity (SIM) scores with any forget sample. These remain samples were used as audio prompts to generate speech with TGU unlearned model. Then, we measured the similarity between the remain sample prompt and the generated output. The results are visualized on 6. A Pearson correlation analysis was conducted to assess the relationship between the similarity of remain samples to forget speakers (x-axis) and the similarity of remain samples to TGU-generated speech (y-axis). Obtained statistic is 0.1396 while the p-value is 0.0003. This indicates a weak positive correlation with statistical significance, meaning that TGU generated speech is generally independent of the remain samples' similarity to forget speakers. Had the model not been robust and mistreated remain samples as forget speaker samples, there would have been a strong negative correlation. Additionally, we found that on remain samples with high similarities with forget speakers (maximum SIM with forget speakers (x-axis) greater than 0.4), the mean of TGU-generated speech similarity (y-axis) is 0.593. This highlights TGU's robustness in handling remain speaker prompts, even when they closely resemble forget speakers.

## H   EXPERIMENT ON GENERAL TASKS

To provide deeper insights on how TGU unlearning may affect model performances on general tasks where ZS-TTS is used, we experiment the original model and TGU on transient noise removal.

Table 7: Transient noise removal results on LibriSpeech test-clean set

| Methods | WER↓ | SIM↑ |
|---|---|---|
| Clean speech | 4.3 | 0.689 |
| Noisy speech | 47.9 | 0.213 |
| Original | **2.4** | **0.666** |
| TGU (proposed) | 2.5 | 0.641 |

Table 8: Diverse speech sampling results on LibriSpeech test-other evaluation set

| Methods | WER↓ | FSD↓ |
|---|---|---|
| Ground truth | 4.5 | 164.4 |
| Original | 8.0 | **170.2** |
| TGU (proposed) | **7.9** | 177.8 |

## H.1 TRANSIENT NOISE REMOVAL

ZS-TTS can be applied in tasks where editing is required to remove undesired noise in speech datasets. To prevent having to go through repetitive and inefficient recording to obtain clean speech, ZS-TTS can generate clean audio for the noisy segment. We follow experimental settings of (Le et al., 2024) to analyze how TGU unlearned model performs on the task of transient noise removal.

From LibriSpeech test-clean dataset samples of durations 4 to 10 seconds, we construct noise at a -10dB signal-to-noise ratio over half of each sample's duration. Table 7 suggests that TGU provides comparable performances to that of the original model. While seemingly low, diminished model performances on transient noise removal is present relatively to the original model. We suggest that this is a trade-off from successful unlearning. While the model has unlearned to generate voice characteristics of the forget dataset, smaller knowledge-base and implemented randomness could have affected its reconstructing abilities.

## H.2 DIVERSE SPEECH SAMPLING

Being able to generate diverse speech is also an important feature of ZS-TTS models as it ensures realistic and high-quality speech that resembles natural distributions. This is necessary in applications such as speech synthesis or generating training data for speech related tasks (e.g., Automatic Speech Recognition). The diversity of generated speech samples is measured with Fréchet Speech Distance (FSD) as suggested in (Le et al., 2024). From generated speech samples, we extracted self-supervised features using 6th layer representation of wav2vec 2.0 (Baevski et al., 2020). The features were reduced to 128 dimensions with principle component analysis and used to calculate the similarity of distributions with real speech. High FSD indicates lower quality and minimal diversity, while low FSD refers to high quality and more diversity. For this experiment, $\alpha$ is set to 0 to ensure more diversity. Ground truth FSD is obtained by partitioning the LibriSpeech test-other set into half while ensuring equal distribution of data per speaker across both subsets

Experimental results in Table 8 show that FSD increases in TGU unlearned model. Because this task does not require input audio prompts, diverse speech sampling relies relatively heavier on datasets used to train the model. Implementing machine unlearning and thus inducing forgetting of specific speakers causes a trade-off in model's diversity. Meanwhile, it is noticeable that TGU achieves a lower WER in this case. We can infer that TGU obtains robustness in relatively noisy dataset comparable to the Original model.

# I  INFERENCE SAMPLES

Figures 7 and 8 show the Mel-spectrograms for the ground truth, original VoiceBox, SGU, and TGU inference results on forget speaker samples. These figures represent samples from speakers *789* and *6821*, respectively. The ground truth Mel-spectrogram corresponds to the audio where the same speaker as the prompt reads the same transcription.

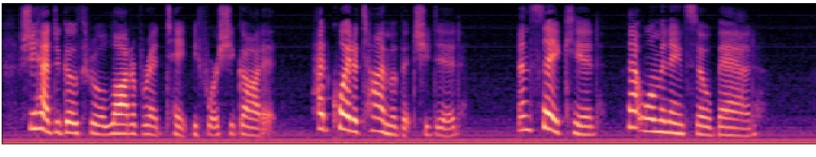

(a) Ground Truth

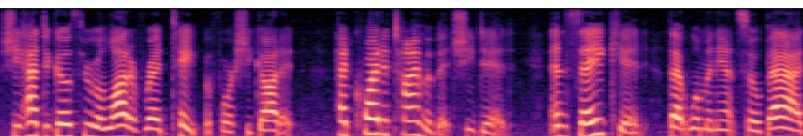

(b) Original

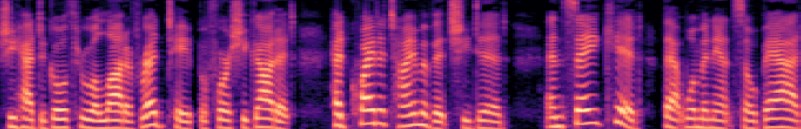

(c) SGU Sample 1

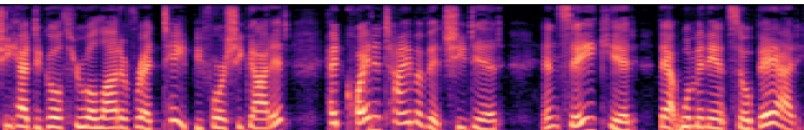

(d) SGU Sample 2

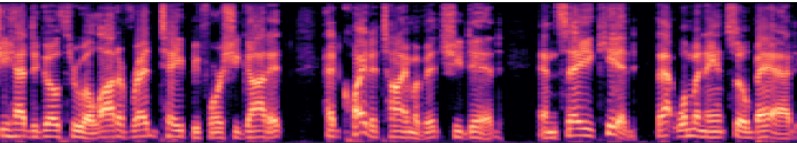

(e) TGU Sample 1

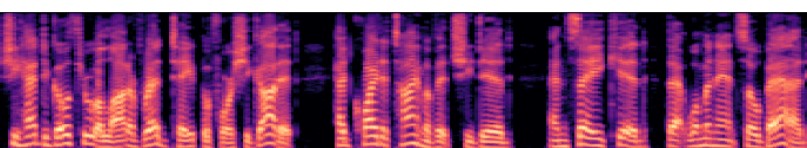

(f) TGU Sample 2

Figure 7: Mel-Spectrogram Comparisons: GT, Original, SGU Samples, and TGU Samples for the forget speaker *789*

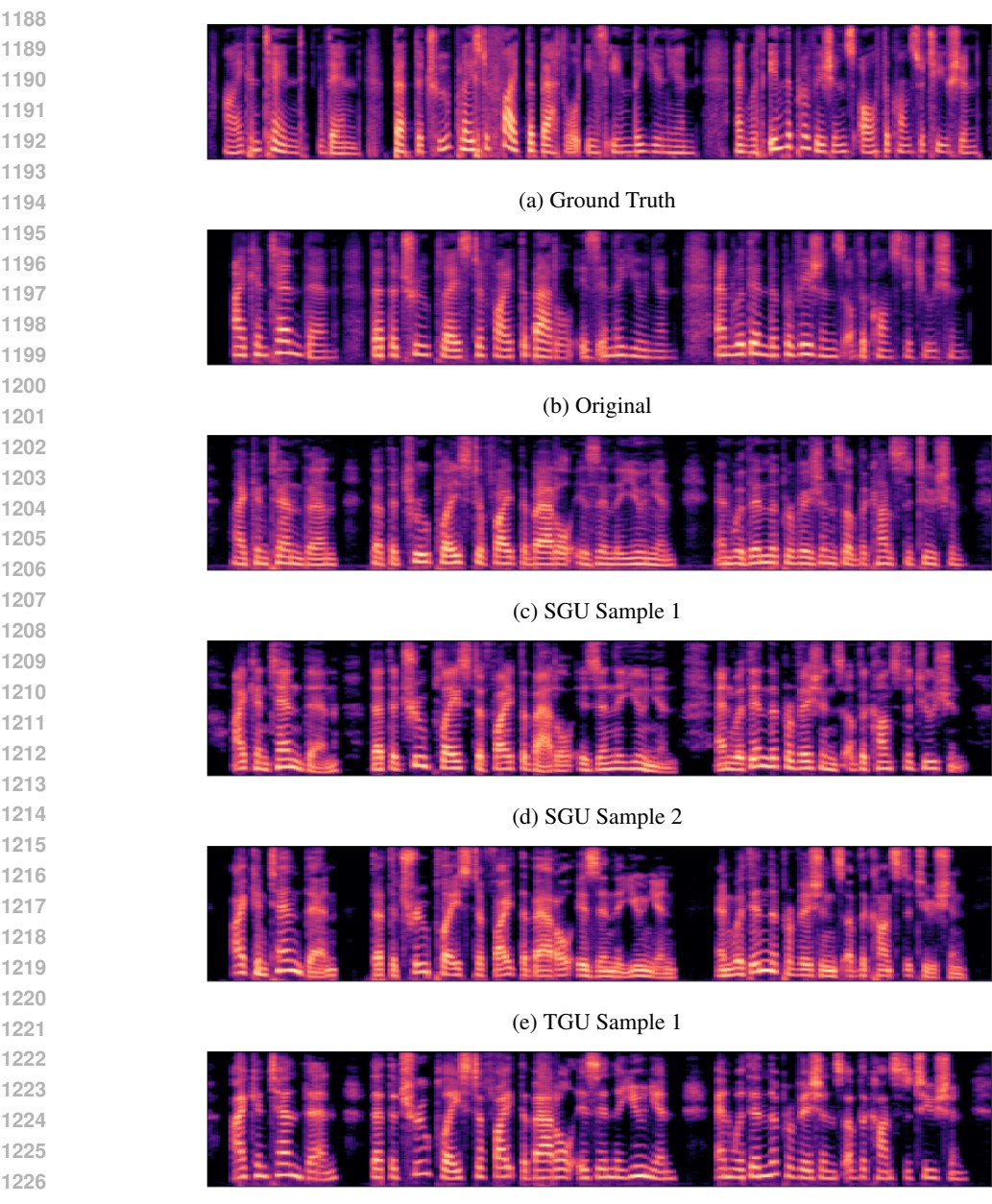

(a) Ground Truth

(b) Original

(c) SGU Sample 1

(d) SGU Sample 2

(e) TGU Sample 1

(f) TGU Sample 2

Figure 8: Mel-Spectrogram Comparisons: GT, Original, SGU Samples, and TGU Samples for the forget speaker *6821*

