# OpenReview forum: "Do Not Mimic My Voice: Teacher-Guided Unlearning for Zero-Shot Text-to-Speech"
_ICLR.cc/2025/Conference — Submitted to ICLR 2025_

### Official Review · Reviewer_98eY · 2024-10-29

**Soundness:** 2
**Presentation:** 2
**Contribution:** 2
**Rating:** 3
**Confidence:** 4

**Summary:**

This paper proposes an unlearning framework for zero-shot text-to-speech (TTS) to prevent the model from replicating the speech characteristics of specific speakers.
1. The authors appear to misunderstand the concept of zero-shot TTS. In Figure 1, it seems that the model functions when the speaker is included in the training set. However, in true zero-shot TTS, speakers should not be present in the training data. This discrepancy undermines the authors’ claim that the framework is suitable for zero-shot TTS.
2. For practical relevance, it would be beneficial to demonstrate the results across multiple models, preferably open-source ones. However, the authors only tested their framework on Voicebox, which is not open-source.
3. The authors acknowledge that model performance degrades significantly as the number of “forgotten” speakers increases, which raises concerns about the practicality of this approach.

**Strengths:**

It seems the authors are first to work on preventing voice clone using machine unlearning.

**Weaknesses:**

1.The models do not work well when forgot speakers number increase. 2. The only model they used are not open-sourced.

**Questions:**

Have the authors try their approach on open sourced models?

---

> ### Author Response · Authors · 2024-11-22
>
> We thank the reviewer for taking the time to review our paper and provide valuable feedback. We hope to clarify some questions and are excited to improve our paper with your valuable feedback.
>
> ### **1. The authors appear to misunderstand the concept of zero-shot TTS. In Figure 1, it seems that the model functions when the speaker is included in the training set.**
>
> Figure 1 seems to contain some ambiguity in its depiction. The figure was included to emphasize the difference of how machine unlearning becomes a more complex task in zero-shot generative tasks.
>
> Machine unlearning addresses the challenge of ensuring a specific knowledge, concept, or dataset can be effectively “forgotten” by a trained model. In non-zero-shot tasks, where the model is explicitly trained on a specific dataset or domain, unlearning a particular data point typically suffices to prevent the generation of that data point during inference. Hence, the ultimate aim is often to suggest an approximate unlearning method as a pathway to efficiently achieve parameters equivalent to exact unlearning (with the retrained model serving as a benchmark).
>
> However in zero-shot tasks, the situation is much more complex. Zero-shot models possess a broad general knowledge base that enable them to perform tasks without explicit exposure to the corresponding training dataset. This means that simply unlearning the data point is insufficient to prevent a model from generating unwanted content as the model’s inherent zero-shot capabilities allow it to reconstruct similar information. This creates a fundamental limitation in achieving “truly unlearned” model parameters, when the goal is to prevent unwanted data. Thus, an exactly unlearned model is nor a benchmark or a golden standard as can be observed in Table 1.
>
> Our figure focuses on this complexity : Current research in machine unlearning often treats retrained models of exact unlearning as the golden standard. However, we address the challenges posed by zero-shot capabilities in large models in this paper.
>
> The VoiceBox model, which we utilized as a base model, is a Zero-Shot TTS model capable of effectively reconstructing voice styles of speakers not seen in training. Based on thorough understanding of ZS-TTS, we would like to refer to statements in L348~350  that states LibriHeavy was used for training, while LibriSpeech-eval (never seen during training) was employed as the Remain Set for evaluation. These two datasets originate from entirely different corpora and do not share overlapping speakers. Thus, the evaluation includes speakers who were not included in the training set.

---

> ### Author Response · Authors · 2024-11-22
>
> ### **2. The only model they used are not open-sourced. Have the authors try their approach on open sourced models?**
>
> To address your question, we chose to experiment on a SOTA closed model, than a low-performance open model.
> There exists models such as YourTTS [1] which adopts a Speaker Encoder-Generation architecture has made its model and training code publicly available as open-source. Unfortunately, due to ethical concerns surrounding potential misuse of the released model (serving as the motivation for our paper), official codes and models for most high-performing ZS-TTS methodologies have not been made publicly accessible.
>
> Thus, significant effort was dedicated to reconstructing a closed source SOTA model to ensure the validity of our experiments and arguments. Our method emphasizes implementation of randomness to remove forget dataset from model parameters and therefore neutralize model’s ability to clone voices. We believe this is an applicable method to be integrated in all ZS-TTS models. We agree it is an essential work to expand our implementation and hope to address this in future work.
>
> [1] Edresson Casanova, Julian Weber, Christopher Shulby, Arnaldo Candido Junior, Eren Gölge, and Moacir Antonelli Ponti. 2022. *YourTTS: Towards Zero-Shot Multi-Speaker TTS and Zero-Shot Voice Conversion for everyone*. In International Conference on Machine Learning, pp. 2709-2720. PMLR, 2022. URL https://arxiv.org/abs/2112.02418.

---

> ### Author Response · Authors · 2024-11-22
>
> ### **3. The authors acknowledge that model performance degrades significantly as the number of “forgotten” speakers increases, which raises concerns about the practicality of this approach. The models do not work well when forgot speakers number increase.**
>
> We understand the concerns on the practicality. The model performance does change depending on the number of “forget” speakers, as described in L517~519 , often causing differences in required training time for desired results.
>
> The experiment was conducted to forget 10 speaker identities (appx. 15 min per speaker) out of train dataset of 6,736 speakers (appx. 50,000 hours in total). An unlearning approach to remove specific identity in image generation [3] forgets 1 identity image at each experiment on a model pre-trained on 70,000 images of human identities. Another research [1] removes 1 specific concept or art style using 1000 images on Stable Diffusion model pre-trained on 5.85 billion image-text pairs for each experiment. Similar to these settings, in practice, the subjects requiring unlearning will likely be few.
>
> The relationship between model performance and the size of forget dataset is a well-observed phenomenon across all domains of machine unlearning. A recent study [2] investigates this as a challenge in machine unlearning methods in Fig.2a and Table A6. Figure 8 of another study [1] also illustrates that increased size of forget dataset comes with significant limitations. This trend parallels the general observations in AI, where models tend to exhibit diminished performances when trained on smaller datasets.
>
> We fully concur with points raised in your assessment, which had served as the motivation for introducing TGU as compared to SGU. While our work initially implemented a more naiive approach SGU, a method which already had obtained a higher unlearning performance than baselines NG and KL, we were inspired to derive main method TGU. TGU focuses on enhancing training efficiency while ensuring improved evaluation under same conditions. Notably, TGU was able to achieve the highest performance in fixed settings faster while others struggled at maintaining model’s retain scores, underscoring our contribution to this domain.
>
> We agree on your concerns and believe this challenge of forget set size and unlearning performance should be addressed in all unlearning domains. In future work, we aim to prioritize achieving faster training times while accommodating larger forget sets.
>
> [1] Rohit Gandikota, Joanna Materzynska, Jaden Fiotto-Kaufman, and David Bau. Erasing concepts
> from diffusion models. In Proceedings of the IEEE/CVF International Conference on Computer
> Vision (ICCV), pp. 2426–2436, 2023. URL https://arxiv.org/pdf/2303.07345.
>
> [2] Chongyu Fan, Jiancheng Liu, Yihua Zhang, Eric Wong, Dennis Wei, and Sijia Liu. Salun: Em-
> powering machine unlearning via gradient-based weight saliency in both image classification and
> generation. In Proceedings of the International Conference on Learning Representations (ICLR), 2024. URL https://arxiv.org/abs/2310.12508.
>
> [3] Juwon Seo, Sung-Hoon Lee, Tae-Young Lee, Seungjun Moon, and Gyeong-Moon Park. Generative unlearning for any identity. In Proceedings of the IEEE/CVF Conference on Computer Vision and  Pattern Recognition (CVPR), pp.9151–9161, 2024.

---

> ### Comment · Reviewer_98eY · 2024-11-25
>
> Thanks to the authors for their response.
>
> Regarding Point 1, while the authors acknowledge the issue with Figure 1, they have made no effort to clarify it. Instead, they provide an extended discussion that is unrelated to my original question and concern.
>
> For Point 2, they state:
>
> "we chose to experiment on a SOTA closed model, than a low-performance open model"
> "There exists models such as YourTTS [1] which adopts a Speaker Encoder-Generation architecture has made its model and training code publicly available as open-source. "
> "Unfortunately, due to ethical concerns surrounding potential misuse of the released model (serving as the motivation for our paper), official codes and models for most high-performing ZS-TTS methodologies have not been made publicly accessible."
>
> This argument is simply inaccurate. Numerous high-quality open-source TTS models are available. However, the authors only refer to a low-performing model from 2021, which is now three years old—and soon to be four.
>
> Overall, the authors' response fails to address my concerns in any meaningful way and instead seems to reconfirm them. As such, I intend to maintain my scores.

---

> > ### Author Response · Authors · 2024-11-25
> >
> > We appreciate the reviewer’s follow-up comment and would like to clarify the availability of open-source Zero-Shot TTS (ZS-TTS) models and our rationale for model selection.
> >
> > Several open-source models, such as YourTTS[1], CosyVoice[2], XTTS[3], and HierTTS++[4], have advanced ZS-TTS by incorporating non-verbal information (e.g., emotion, punctuation) or extending functionality to include voice conversion. However, these models rely on x-vector or speaker encoder-based architectures, which align more closely with classification-based systems. As a result, privacy concerns in these models could potentially be mitigated using existing classification-based machine unlearning algorithms explored in prior research.
> >
> > In contrast, in-context learning-based methods like VoiceBox [5], VALL-E [6], and NaturalSpeech 3 [7] achieve superior speaker style replication from minimal input but pose greater challenges for unlearning due to their generative nature. Unfortunately, official implementations of these models are not publicly available, and unofficial versions lack fidelity guarantees. Given these constraints, we selected VoiceBox for its state-of-the-art capabilities and suitability for evaluating our proposed framework.
> >
> > We also acknowledge the recent release of MaskGCT [8] and F5TTS [9] in late 2024. While promising, these models became available too recently (and were submitted to ICLR 2025 alongside our work) to be included in this study.
> >
> > We hope this clarification addresses the reviewer’s concerns.
> >
> > [1]Edresson Casanova, Julian Weber, Christopher Shulby, Arnaldo Candido Junior, Eren Gölge, and Moacir Antonelli Ponti. 2022. *YourTTS: Towards Zero-Shot Multi-Speaker TTS and Zero-Shot Voice Conversion for everyone*. In International Conference on Machine Learning, pp. 2709-2720. PMLR, 2022.
> >
> > [2]Du, Zhihao, et al. "Cosyvoice: A scalable multilingual zero-shot text-to-speech synthesizer based on supervised semantic tokens." *arXiv preprint arXiv:2407.05407* (2024).
> >
> > [3]Casanova, Edresson, et al. "XTTS: a Massively Multilingual Zero-Shot Text-to-Speech Model." *arXiv preprint arXiv:2406.04904* (2024).
> >
> > [4]Lee, Sang-Hoon, et al. "Hierspeech++: Bridging the gap between semantic and acoustic representation of speech by hierarchical variational inference for zero-shot speech synthesis." *arXiv preprint arXiv:2311.12454* (2023).
> >
> > [5] Le, Matthew, et al. "Voicebox: Text-guided multilingual universal speech generation at scale." *Advances in neural information processing systems* 36 (2024).
> >
> > [6] Wang, Chengyi, et al. "Neural codec language models are zero-shot text to speech synthesizers." *arXiv preprint arXiv:2301.02111* (2023).
> >
> > [7] Ju, Zeqian, et al. "Naturalspeech 3: Zero-shot speech synthesis with factorized codec and diffusion models." *arXiv preprint arXiv:2403.03100* (2024).
> >
> > [8] Wang, Yuancheng, et al. "Maskgct: Zero-shot text-to-speech with masked generative codec transformer." *arXiv preprint arXiv:2409.00750* (2024).
> >
> > [9] Chen, Yushen, et al. "F5-tts: A fairytaler that fakes fluent and faithful speech with flow matching." *arXiv preprint arXiv:2410.06885* (2024).

---

> > ### Author Response · Authors · 2024-11-27
> >
> > Regarding point 1, we wanted to reassure the reviewer about how we understood the concept of zero-shot TTS and thus provided following discussion. We are sorry as it seems we have not fully addressed your main concerns.
> >
> > To fully address this issue, we revised figure 1 accordingly in the revised paper. We believe the discrepancy came from depicting a training process and an inference process in one figure. It has come to our eyes that this figure is ambiguous in addressing the natures of zero-shot TTS models and have revised our paper accordingly.
> >
> > With respect, we would like to ask the reviewer if their concerns have been addressed on the updated figure. Thank you for strengthening our paper with your valuable feedback.

---

### Official Review · Reviewer_zTeM · 2024-10-29

**Soundness:** 3
**Presentation:** 3
**Contribution:** 3
**Rating:** 8
**Confidence:** 4

**Summary:**

The paper introduces a novel Teacher-Guided Unlearning (TGU) framework, which allows models to forget specific speaker identities while retaining the ability to synthesize speech for other speakers. This is particularly relevant given the potential misuse of ZS-TTS systems that can replicate voices without consent. The proposed method is built on top of VoiceBox (Le et al. 2024, from Meta) which has reached the SOTA as a ZS-TTS model.

**Strengths:**

The key strengths of the paper include:
Privacy Concerns: The rapid development of ZS-TTS raises ethical issues, particularly the unauthorized replication of individuals' voices, necessitating effective machine unlearning techniques to protect voice privacy.
TGU Framework: Proposed TGU is the first machine unlearning framework specifically designed for ZS-TTS. It utilizes a pre-trained teacher model to guide the generation of speaker-randomized outputs, effectively helping the model to forget specific speaker identities while maintaining performance for others.
Randomness in Outputs: Unlike traditional unlearning methods, TGU incorporates randomness in voice styles when the model encounters prompts related to forgotten speakers, which helps neutralize the model's responses to these prompts.
Evaluation Metrics: The paper introduces a new evaluation metric, speaker-Zero Retrain Forgetting (spk-ZRF), to measure the effectiveness of the unlearning process. The results indicate that TGU not only limits the replication of forgotten voices but also preserves the quality of speech generation for remaining speakers.

**Weaknesses:**

Evaluation Metrics: The introduction of spk-ZRF is a valuable contribution, as it provides a quantitative measure of the unlearning effectiveness. However, the paper could benefit from a more detailed explanation of how this metric compares to existing metrics in the literature.

Randomness Implementation: The paper emphasizes the importance of randomness in unlearning, yet it does not sufficiently address potential trade-offs between randomness and speech quality. The balance between generating random outputs for forget speakers while maintaining high fidelity for others needs further exploration.

Complexity of Implementation: The introduction of randomness may complicate the training process and could lead to inconsistent performance across different applications. A clearer discussion on how to balance randomness with quality would be beneficial.

Limited Scope of Forgetting: The focus on only preventing replication of specific voices may overlook broader implications, such as how unlearning affects overall model performance or its ability to generalize across different tasks. A more holistic approach could provide deeper insights into the trade-offs involved.

Dataset size: Used Dataset size is relatively small, may not representative of practical scenarios.

**Questions:**

1. Perform a computational analysis detailing computational costs, training time requirements, comparison of computational overhead with
baseline approaches, and inference time and resource requirements.

---

> ### Author Response · Authors · 2024-11-22
>
> We thank the reviewer for taking the time to review our paper and provide valuable feedback. We truly appreciate this opportunity to strengthen our paper with your insights.
>
> ### **Q1. Perform a computational analysis detailing computational costs, training time requirements, comparison of computational overhead with baseline approaches, and inference time and resource requirements.**
>
> **Compuational Resources**
>
> We would like to provide additional details regarding the computational analysis of our study. VoiceBox has model parameter size of approximately 328M.
>
> |  | GPU (#) | Total Training Time | Real Time Factor (RTF) on A100(40GB) | # of Steps | Seconds per Iteration |  Forget Batch Size | Remain Batch Size |
> |------|-------|------|------|------|------|------|------|
> |Original Pretrained Model| A100(40GB) 8 | 100 hrs (appx.) | 0.71 | 500K | 1.68 | - | - |
> |NG| A100(40GB) 1 | 100 hrs (appx.) | 0.71 | 9.5K | 8.08 | 64 | - |
> |KL| A100(40GB) 1 | 187 hrs (appx.) | 0.71 | 32.5K | 16.56 | 8 | 32 |
> |SGU| A100(40GB) 8 | 75 hrs (appx.) | 0.71 | 145K | 2.68 | 8 | 32 |
> |TGU| A100(40GB) 8 | 250 hrs (appx.) | 0.71 | 145K | 7.21 | 8 | 32 |
>
>
> **Training Duration and Performance Considerations**
>
> The total training time is shorter in NG and KL only due to the fact they were intentionally halted at 9,500 and 32,500 steps, respectively. This decision was based on our observations that further training led to diminishing returns in terms of model performance and effectiveness in unlearning, with both models struggling to maintain performance beyond these points.
>
> Additionally, it is pertinent to mention that the KL and TGU methods incorporate a teacher model. This inherently extends the training time per step due to the additional computations required.
>
> **Inference**
>
> Regarding inference, we note that the unlearning methods implemented do not influence the size of the final unlearned models. Consequently, the inference time per sample remains consistent across all methods -0.71 RTF on A100(40GB).
>
> We trust that these clarifications will assist in the thorough evaluation of our work. Thank you for your attention to these details, we have realize the necessity of this analysis in our revision of final paper.

---

> > ### Comment · Reviewer_zTeM · 2024-11-29
> >
> > Thank you for adding the details of the computational cost. Could you please include inference time details in the revised version for the sample outcomes mentioned on the preview website https://speechunlearn.github.io? While the whole process is computationally expensive, this work shows promise as a new contribution to the field, though I have concerns about its practical applications due to the computational demands.

---

> ### Author Response · Authors · 2024-11-22
>
> ### **W1. Evaluation Metrics: The introduction of spk-ZRF is a valuable contribution, as it provides a quantitative measure of the unlearning effectiveness. However, the paper could benefit from a more detailed explanation of how this metric compares to existing metrics in the literature.**
>
> We sincerely appreciate your observation. The motivation for proposing spk-ZRF is spread out across the paper, and we will ensure that this aspect is more clearly articulated in the revised version. To elaborate on L282-283, the primary motivation for introducing spk-ZRF stems from the absence of a sufficient existing metric.
>
> Commonly used methods, such as Completeness, JS-Divergence, Activation Distance, and Layer-wise Distance, focus on calculating the distance between behaviors on the remain set and the forget set. These approaches, as exemplified in our Table 1, compare the model performance of the unlearned model with the original model. While low performance on the forget set and bigger gap between remain and forget set is often used as a measure of unlearning success, this alone does not necessarily confirm whether the model has truly removed the forget set's information. For instance, the scenario described in L 222–226 highlights an insufficient unlearning method where a forgotten speaker's voice can be easily reconstructed, yet the model may appear to be well-forgotten if evaluated solely on existing metrics.
>
> Epistemic Uncertainty, another existing metric used in the unlearning domain [[1](https://arxiv.org/pdf/2208.10836)], evaluates how little information about the forget dataset in the unlearned model. However, applying this metric on ZS-TTS models is not suitable as the representations in model layers contain deeply entangled information about speaker identity and the knowledge to generate audible speech from a given text transcript. Consequently, low epistemic uncertainty does not necessarily indicate that the model has successfully forgotten speaker-specific information.
>
> Our focus was to design an evaluation metric **specifically targeting the removal of speaker identity information while isolating this from the performance of audible speech generation**. For these reasons, we developed spk-ZRF as suggested in this paper, intended to complement existing methods and provide a more targeted approach to evaluation. Thank you once again for your valuable feedback, which has helped us refine our presentation of this concept.
>
> [1] Alexander Becker and Thomas Liebig. 2022. *Evaluating Machine Unlearning via Epistemic Uncertainty*. URL https://arxiv.org/abs/2208.10836.

---

> > ### Comment · Reviewer_zTeM · 2024-11-30
> >
> > Thank you for your detailed response regarding the spk-ZRF metric and its motivation. I appreciate your acknowledgment of my comment and your commitment to clarifying the rationale behind this new metric in the revised version of your paper.
> > As you refine your presentation of spk-ZRF, I would suggest including a comparative analysis in the revised paper that explicitly outlines how spk-ZRF differs from and improves upon these existing metrics. This could further solidify its place in the literature and clarify its unique contributions to the field of machine unlearning in zero-shot text-to-speech systems.

---

> ### Author Response · Authors · 2024-11-28
>
> ### **W.3-4 Complexity of Implementation and Limited Scope of Forgetting**
>
> **The focus on only preventing replication of specific voices may overlook broader implications, such as how unlearning affects overall model performance or its ability to generalize across different tasks. A more holistic approach could provide deeper insights into the trade-offs involved.**
>
> We agree that our experiments focused on replication of voice styles, despite ZS-TTS having more applications. To address potential trade-off in model's ability across different tasks and applications, we compare the unlearned model's performance on two tasks : (1) transient noise removal and (2) diverse speech sampling.
>
> **(1) Transient noise removal**
>
> ||WER|SIM|
> |------|------|------|
> |Clean Speech|4.3|0.689|
> |Noisy Speech|47.9|0.213|
> |Original|2.4|0.666|
> |TGU|2.5|0.641|
>
> In real world, ZS-TTS can be used to edit audio. For instance, if an actor were to misspeak a word or unintended noise was introduced during recording, ZS-TTS could be applied to generate clean audio. For this experiment, we purposely construct a speech with noise at a -10dB signal-to-noise ratio over half of each sample’s duration from LibriSpeech test-clean. Then, we prompted the model to generates clean speech for the corresponding noisy segment.
>
> Based on WER and SIM, we can note that the TGU model shows decrease in model performance in reconstructing speech for noisy segments. This performance gap is quite similar to the gap observable in Table 1 of our paper. This is because the model is able to fully mask over the noisy segment and predict to infill with clean generated output. For these tasks, WER is an imperative metric as correct content should be generated to edit audios. However from both Table 1 and this experiment, WER seems to increase as randomness is implemented. While Exact Unlearning and Fine Tuning methods do not reach sufficient unlearning, they are able to maintain low WER scores whereas randomness increases WER with SGU and TGU.
>
> **(2) Diverse speech sampling**
>
> ||WER|SIM|
> |------|------|------|
> |Ground Truth|4.5|164.4|
> |Original|8.0|170.2|
> |TGU|7.9|177.8|
>
> We also explore diverse speech sampling to measure whether the unlearned model maintains to synthesize different voice styles. Being able to generate diverse speech is also an important feature of ZS-TTS models as it ensures realistic and high-quality speech that resembles natural distributions. This is necessary in applications such as speech synthesis or generating training data for speech related tasks (e.g., Automatic Speech Recognition). The diversity of generated speech samples is measured with Fr´echet Speech
> Distance (FSD) as suggested in [1]. High FSD indicates lower quality and minimal diversity, while low FSD refers to high quality and more diversity. The experiment is conducted by giving only the text prompt, and no audio prompt of LibriSpeech test-other. Even with no audio prompt, the model should be able to generate diverse and natural speech with distribution similar to real samples. Further implementation details are in Appendix H.2 of revised paper.
>
> In this experiment, TGU unlearned model shows worse quality and diversity distribution in generated speech than the original model. While TGU robustly handles noisier LibriSpeech test-other dataset with WER lower than Original model, the lacking diversity could affect future applications of ZS-TTS. Ideally, removal of specific knowledge should be performed with minimal changes in model's output distribution.
>
> -----------------
>
> Implementing randomness for machine unlearning may result in a degradation of model performance in terms of both content generation accuracy and diversity. We attribute the observed lower performance in WER to the reliance on model-generated targets as guidance in SGU and TGU. If these targets are not generated accurately beforehand, the model is likely to learn erroneous patterns. Additionally, the reduction in diversity appears to be a consequence of removing the influence of the forget dataset, as this effectively reduces the model's overall knowledge. Future work could focus on preserving the diversity of speaker styles while effectively mitigating unwanted generation, potentially through disentangled representations. While we were able to maintain a comparable level of performance and diversity, the trade-offs are likely to intensify as the size of the forget dataset increases.
>
> We thank the reviewer for strengthening our research and would like to note that we have added these experiments in the revised version. Please let us know if we have addressed your concerns, we look forward to your answer.
>
> [1] Matthew Le, Apoorv Vyas, Bowen Shi, Brian Karrer, Leda Sari, Rashel Moritz, Mary Williamson, Vimal Manohar, Yossi Adi, Jay Mahadeokar, et al. Voicebox: Text-guided multilingual universal speech generation at scale. Advances in neural information processing systems, 36, 2024.

---

### Official Review · Reviewer_ef5p · 2024-11-04

**Soundness:** 4
**Presentation:** 2
**Contribution:** 1
**Rating:** 3
**Confidence:** 4

**Summary:**

The paper proposes a new problem to address for the zero-shot TTS model, which is to unlearn an existing voice. The authors provide a simple solution, which is to fine-tune the model on the original training set along with a newly defined target generated by the original teacher model without the target speaker as the reference. As a result, the fine-tuned model keeps the original performance on other speakers while generating random speaker identities for the selected speakers whose voices are supposed to be removed.

**Strengths:**

* **Originality**: This work is the first in the field to define voice unlearning for zero-shot TTS (ZS-TTS) and propose a simple solution with synthetic data to address it. It has also proposed a new metric, spk-ZRF, to examine the degree of reconstructability of the target speaker that is supposed to be unlearned.

* **Quality**: The paper has compared several baselines with various metrics and demonstrated the effectiveness of this proposed method. It also has a nice visualization to showcase the effects of various methods for achieving this goal.

* **Clarity**: The presentation of the paper is fairly clear, with all necessary symbols defined, which made it not difficult to follow.

* **Significance**: It is the first paper in ZS-TTS to address the voice unlearning problem with a new metric to account for the reconstructability of the target speaker that can have a significant influence on future works in this field.

**Weaknesses:**

The major weakness of this work is its use case is unclear. The problem being solved is not well-motivated, especially in a non-zero-shot setting. Under what condition would the user (in this case, the model trainer or machine learning engineer) re-train the model with all the training data to remove some voices? What benefits does this method provide? It is not obvious to come up with a practical use case for the proposed method where the entire training data is needed to fine-tuned the model for 145k steps (more than a quarter of the 500k steps of the base model training), and the specific speaker that needs to be forgotten has to come with at least 5 minutes of their training audio. In fact, some recent work [1] in image generation has provided more interesting methods for zero-shot unlearning, where only a single image is needed for the model to stop generating the target facial identity.

Since this method requires the entire training data of the original model, 5 minutes of audio for the forget speaker, and more than a quarter of training iterations of the original model, the actual significance of this work is rather limited. For example, in the case voice unlearning is to be used by a cloud service provide, if a zero-shot TTS service provider wants to prevent the model from cloning certain voices, they can easily use a speaker verification model to check whether the provided prompt speaker collides with a database of speaker embeddings whose voices are not supposed to be used and stop providing the service if the provided voice is in the forbidden speaker database. On the other hand, if it is for an open-source model, it is also possible to fine-tune the model on some other dataset for the model to regain the ability to clone the forgotten speaker's voice. From the paper, it is unclear how much data is needed to recover the forgotten speaker's voice as the paper does not show it. The significance of this work could be higher if the proposed method requires an enormous amount of data for the unlearned model to regain its ability to reproduce the voice. However, since no experiment has been conducted, it is unknown whether this work would benefit the open-source community either.

Due to these reasons, the significance of this paper is limited in its current state. It is suggested that the authors provide more motivations and practical use cases of the proposed method and the initial problems of unlearning certain voices to begin with, as it is a new problem proposed by this paper, and so far, the problem does not seem to be very meaningful, and the proposed solution makes it even less effective in practice.


[1] Seo, J., Lee, S. H., Lee, T. Y., Moon, S., & Park, G. M. (2024). Generative Unlearning for Any Identity. In Proceedings of the IEEE/CVF Conference on Computer Vision and Pattern Recognition (pp. 9151-9161).

**Questions:**

1. What is the intended use case of this unlearning scheme?

2. How much data is needed to recover the unlearned speech in both the non-zero-shot setting (where the forget speaker is in the training set) and the zero-shot setting (where the forget speaker is not in the training set)?

**Details Of Ethics Concerns:**

There is no information about the demographics, compensation, or criteria for hiring human subjects for the subjective evaluation. Please add this information to the appendix. Also, please indicate whether you have obtained any IRB approval for these evaluations.

---

> ### Author Response · Authors · 2024-11-22
>
> Thank you for your insightful comments and queries regarding the motivations and practical implications of our approach, particularly concerning the necessity to retrain models to remove certain voices. We appreciate your perspective and acknowledge the need to clarify these points more comprehensively.
>
> ### **1. What is the intended use case of this unlearning scheme?**
>
> **Under what condition would the user (in this case, the model trainer or machine learning engineer) re-train the model with all the training data to remove some voices? What benefits does this method provide?**
>
> In response to your concerns about the conditions under which one would re-train a model to ensure voice privacy, we emphasize the current challenges within the ZS-TTS domain. The majority of ZS-TTS models are kept private, often lacking in publicly available code implementations or pre-trained models due to developers' legitimate concerns over potential malicious use [1]. This secrecy stems largely from privacy considerations, necessitating stringent safeguards before releasing models openly. We also introduce a real life case where a speech generating model led to a lawsuit due to speaker cloning issue of a certain celebrity [2]. We confidently state that methods to prevent model misusage contributed greatly to our motivation of this work. Our proposed method of unlearning certain voice data is actually a necessary step to enhance the practical utility and safety of ZS-TTS technologies.
>
> **Since this method requires the entire training data of the original model, 5 minutes of audio for the forget speaker, and more than a quarter of training iterations of the original model, the actual significance of this work is rather limited.**
>
> | Unlearning Method | # Steps | WER-R | SIM-R | WER-F | SIM-F |
> |------|------|------|------|------|------|
> |SGU|58 K|2.6|0.522|2.6|0.179|
> |SGU|72.5 K|2.6|0.581|2.6|0.377|
> |SGU|145 K|2.6|0.523|2.5|0.194|
> |TGU|58 K|2.8|0.568|2.7|0.203|
> |TGU|72.5 K|2.7|0.588|2.6|0.174|
> |TGU|145 K|2.5|0.631|2.4|0.169|
>
> **Step size**
>
> Our decision to use 145,000 steps is that this constitutes a sufficiently modest number of steps for fine-tuning a Flow-Matching Audio Generation model. In the context of related research, the AudioBox paper [[3](https://arxiv.org/abs/2312.15821)] employs approximately 200,000 steps to finetune a Flow-Matching SSL model for a Zero-Shot TTS task. SpeechFlow [[4](https://arxiv.org/abs/2310.16338)] uses 145,000 steps for finetuning a Flow-Matching SSL model. We decided to employ the smaller number of steps 145,000 with regards to the fact that it is considered a modest step size in domain of Zero-Shot TTS.  The steps in NG and KL are relatively small only due to the fact they were intentionally halted at 9,500 and 32,500 steps, respectively. With further training, these methods worsened model performance and we plan to update the results for comparison in following days.
>
> **Training dataset**
>
> Regarding the utilization of the remain set for implementing unlearning in TGU, we regret any ambiguity caused by our initial presentation. To clarify, TGU does not utilize the full dataset originally used for training; they are trained on approximately 50% of the data. Additionally, as noted in the appendix D of our paper, our methods achieved convergence at around 60,000 steps, utilizing only 21% of the total dataset, contrary to the 145,000 steps mentioned. Our decision to use 145,000 steps was intended to constrain experimental settings across all methods.
>
> **Forget speaker audio duration**
>
> We thank the reviewer for the opportunity of extensive experiments that will strengthen our paper, and plan to update experimental results using a shorter audio duration of forget speakers during this period as well.
>
> **Significance of our work with domain-specific challenges**
>
> While it would be most ideal to achieve a level of unlearning in smaller iterations and dataset size, this study makes an important contribution by achieving unlearning in ZS-TTS, which is particularly intricate. From a given audio and transcribed text, ZS-TTS models learn not only to imitate that speaker identity, but also to convert text to spoken words at the same time. Thus a major challenge lies in selectively unlearning only the speaker identity while maintaining the capability to accurately vocalize text from the forget set as our objective in L209-219. A recent study [[3](https://arxiv.org/abs/2406.01257)] explores challenge of unlearning while the forget and remain set share indistinguishable representations (here, knowledge to vocalize from given text prompt). Shared knowledge in forget and remain set can easily cause unlearning in remain set as well - as depicted in baselines NG and KL. For example, if a forget speaker utters “I am happy”, we have a challenge to forget only the speaker identity, not the ability to correctly vocalize the text “I am happy”.

---

> > ### Author Response · Authors · 2024-11-22
> >
> > This study’s exploration is a significant contribution as we have effectively unlearned speaker identity apart from the ability to vocalize text with a partial of training dataset. We can enhance the safety and utility of ZS-TTS applications, ensuring they serve beneficial purposes without compromising individual privacy.
> >
> > [1] Mackenzie Tatananni. Microsoft AI That Clones Voices to Sound ‘human’ Can’t Be Released to Public. The US Sun, 2024. [www.the-sun.com/tech/11883067/microsoft-ai-voice-clone-tts-system-public-release/](http://www.the-sun.com/tech/11883067/microsoft-ai-voice-clone-tts-system-public-release/).
> >
> > [2] Nick Robins-Early. CHATGPT Suspends Scarlett Johansson-like Voice as Actor Speaks out against OpenAI. The Guardian, 2024. [www.theguardian.com/technology/article/2024/may/20/chatgpt-scarlett-johansson-voice](http://www.theguardian.com/technology/article/2024/may/20/chatgpt-scarlett-johansson-voice).
> >
> > [3] Apoorv Vyas, Bowen Shi, Matthew Le, Andros Tjandra, Yi-Chiao Wu, Baishan Guo, Jiemin Zhang, Xinyue Zhang, Robert Adkins, William Ngan, Jeff Wang, Ivan Cruz, Bapi Akula, Akinniyi Akinyemi, Brian Ellis, Rashel Moritz, Yael Yungster, Alice Rakotoarison, Liang Tan, Chris Summers, Carleigh Wood, Joshua Lane, Mary Williamson, and Wei-Ning Hsu. Audiobox: Unified Audio Generation with Natural Language Prompts. 2023. URL https://arxiv.org/abs/2312.15821.
> >
> > [4] Alexander H. Liu, Matt Le, Apoorv Vyas, Bowen Shi, Andros Tjandra, and Wei-Ning Hsu. Generative Pre-training for Speech with Flow Matching. 2023. URL https://arxiv.org/abs/2310.16338
> >
> > [5] Kairan Zhao, Meghdad Kurmanji, George-Octavian Bărbulescu, Eleni Triantafillou, and Peter Triantafillou. What makes unlearning hard and what to do about it. In NeurIPS 2024. URL https://arxiv.org/abs/2406.01257.

---

> > ### Comment · Reviewer_ef5p · 2024-11-23
> >
> > I appreciate the authors' responses. However, I believe the responses did not address any of my concerns. The authors still could not justify the limitations in the training process: the model still requires 5 minutes of target forget speaker, a quarter of training iteration, and (after the authors clarified the misunderstanding) 50% of the original training data to achieve this effect.
> >
> > The authors made an argument to state the importance of forgetting the voice on the service provider side. However, retaining a speaker embedding is still better than obtaining 5 minutes of the user data just to train the model to forget its voice in terms of privacy concerns. Moreover, given the difficulty of using this technique, it is impractical to do mass forgetting with voices on demand, making it unsuitable to be used for service providers.

---

> ### Author Response · Authors · 2024-11-22
>
> ### **2. If a zero-shot TTS service provider wants to prevent the model from cloning certain voices, they can easily use a speaker verification model to check whether the provided prompt speaker collides with a database of speaker embeddings whose voices are not supposed to be used and stop providing the service if the provided voice is in the forbidden speaker database.**
>
> We appreciate the reviewer’s suggestion regarding the use of a speaker verification model as a mechanism to prevent the unauthorized use of certain voices in a zero-shot TTS service. While this approach provides a practical barrier to unauthorized usage, we argue that it does not fulfill the core principle of the “Right to be Forgotten.” Blocking the generation of the forbidden speaker’s voice does not necessarily imply that the model has truly “forgotten” the speaker.
>
> Additionally, implementing such a system would require the service provider to retain a database of embeddings for the forbidden speakers, contradicting the notion of fully respecting the privacy of individuals who request their voice to be forgotten. Moreover, this approach introduces a significant security vulnerability; if the model was to be compromised, an attacker could exploit the system to replicate the voice of the forbidden speakers.
>
> Our proposed unlearning framework aims to address these issues by ensuring that the model itself forgets the speakers at a fundamental level, rendering it incapable of generating that speaker’s voice, even under adversarial conditions. This not only respects user privacy but also mitigates potential risks associated with model exploitation.

---

> ### Author Response · Authors · 2024-12-04
>
> ### **3. Since this method requires the entire training data of the original model, 5 minutes of audio for the forget speaker, and more than a quarter of training iterations of the original model, the actual significance of this work is rather limited.**
>
> |Method|Steps|Total Duration of each Forget Speakers|WER-R|SIM-R|WER-F|SIM-F|
> |------|------|------|------|------|------|------|
> |TGU|145 K| 15 min| 2.5| 0.631| 2.4 | 0.169|
> |TGU|48 K| 15 min| 2.7 | 0.481| 2.7 | 0.198|
> |TGU w/ Augmentation| 48 K | 1 min | 2.6| 0.612| 2.6| 0.334|
>
> We express our gratitude to the reviewer for allowing us to explore practical applications with shorter audio durations.
>
> In this figure, we illustrate the performance of unlearned model originally reported in the paper at two checkpoints : 145 K steps and 48 K steps. Additionally, we include an experiment where only an average of 1 minute of audio per speaker is used for the unlearning process. This experiment leverages the original ZS-TTS model to augment 1 minute of audio. Using this 1-minute audio as a prompt, we generate diverse speech samples during training to mitigate overfitting to specific transcriptions. With such a limited amount of audio,it is highly unlikely that the model will unlearn only specific speaker characteristics. The model is prone to overfitting on the unlearning dataset, which could result in forgetting only specific utterances spoken by the forget speaker. Consequently, the model may fail to generalize its forgetting to the forget speaker uttering a speech not seen in the unlearning train set. To address this, we implement data augmentation to enhance the effectiveness of unlearning.
>
> Our results indicate that the unlearning is more successful when provided with longer duration of forget speaker’s audio. This is likely because using a limited set of audio prompt, even with augmentation, can only generate a constrained range of styles (rather than being able to portray forget speaker’s characteristics in different emotions, accents, rhythms). We agree that exploring an approach using less datasets per forget speaker would be very practical and thank the reviewer for providing an insight from views of industrial application.

---

> ### Author Response · Authors · 2024-12-04
>
> ### **4. How much data is needed to recover the unlearned speech in both the non-zero-shot setting (where the forget speaker is in the training set) and the zero-shot setting (where the forget speaker is not in the training set)?**
>
> Thank you for your suggestions of recover experiments. For recovering experiments, we halted at early steps because of significantly low model performances. Also, our proposed paper's experiment has not yet extended to unlearn speech on an out-of-domain forget set, so we present the following experiment.
>
> |Methods|Unlearned Steps| Forget Audio Duration for Recover Training|Recover Steps|WER-R|SIM-R|WER-F|SIM-F|
> |----|----|----|----|----|----|----|----|
> |TGU|145 K|-|-|2.5|0.631|2.4|0.169|
> |TGU|145 K|15 min|36.25 K (25% of Unlearned Steps)|4.23|0.303|2.5|0.735|
> |TGU|145 K|1 min| 14.5 K (10% of Unlearned Steps)|4.61|0.226|2.8|0.162|
>
> Aligning with our motivation to make ZS-TTS models safe, we presume a scenario of a privacy attacker who attempts to retrieve the original model parameters. We train the unlearned checkpoints on all 10 of forget speaker’s dataset to recover the original model. We also presume a practical scenario as the reviewer has highlighted important, and  attempt to recover the model performance using average of 1 minute for each speaker.
>
> Given a long audio duration of 15 minutes for the forget speakers, we can note that the model has become overfitted to generate speech specifically mimicking the forget speaker’s voice. The recovered model fails to retrieve its original model’s zero-shot capabilities, and is more likely to generate wrong speech content with higher WER. This process resembles fine-tuning a zero-shot model for specific speakers rather than true recovery. Consequently, the original ZS-TTS model cannot be restored, and the attacker is essentially leveraging transfer learning to create a forget speaker-specific TTS model, provided sufficient training data for the forget speaker is available. However, with enough training data, the attacker could achieve similar results using any other non-zero-shot TTS model.
> Taking the reviewer's emphasis on practicality into account, we also consider a scenario where an attacker has access to 1 minute of the forget speaker’s voice sample. In this case, the model parameters also remain unrecoverable and the model fails to generate forget speaker’s voice. The model loses its zero-shot abilities hence the performance at early steps. Therefore, in practical scenarios where an attacker may attempt to train the model to clone an individual’s voice with short sample of speech  (e.g., voice phishing), it would not be feasible to recover the model or successfully generate the forget speaker’s voice.
>
> As previously discussed, current ZS-TTS models are often closed to prevent malicious usage. This issue is particularly critical given that these models have achieved notable performance, yet most remain inaccessible for open use. Therefore, prioritizing safety and privacy is imperative to enable widespread and responsible accessibility of these technologies. With the TGU unlearned model, it would not be possible for any user to recover the original ZS-TTS model unless the service provider explicitly grants open access to its training dataset. While this approach may not fully eliminate the risks associated with cloning individual voices in scenarios where extensive datasets are available, it represents a meaningful step toward enhancing the safety and ethical use of such advanced technologies.

---

### Official Review · Reviewer_r7jW · 2024-11-05

**Soundness:** 3
**Presentation:** 2
**Contribution:** 2
**Rating:** 5
**Confidence:** 3

**Summary:**

The paper introduces TGU (Teacher-Guided Unlearning), a novel mechanism that enhances zero-shot text-to-speech (TTS) systems to address ethical and privacy concerns. To effectively assess the speaker randomness of the 'forget prompts' used within the system, the authors have developed a new metric named spk-ZRF. The experimental findings presented in the study validate the effectiveness of the proposed TGU framework, showcasing its potential to significantly improve the safety and privacy aspects of TTS technologies.

**Strengths:**

1. The paper is rooted in a meaningful motivation. The advancement of zero-shot TTS systems raises ethical and privacy concerns and highlights potential misuse for fraudulent activities. This relevance to real-world issues underscores the importance of the research.
2. The introduction of the novel metric spk-ZRF for evaluating speaker randomness in 'forget prompts' is a commendable aspect of the paper. This new metric contributes to the field by providing a quantifiable measure to assess the efficacy of the proposed unlearning mechanism.
3. The clarity and quality of the figures presented in the document significantly aid in understanding the complex concepts and methodologies described, making the results and processes more accessible and comprehensible to the audience.

**Weaknesses:**

1. The "fine-tune" process of TGU is based on given forgotten prompts, which naturally raises a question regarding how this framework performs with other prompts from the same forgotten speaker. Moreover, it is unclear how the framework handles prompts from other retained speakers who have similar timbres with a forgotten one. The paper currently lacks a discussion on these aspects, which are crucial for comprehensively evaluating the robustness and versatility of the proposed TGU framework.
2. The TGU mechanism resembles a distillation process, suggesting another potential baseline worth exploring. The approach could involve using the zero-shot TTS model to generate audio with the retained speaker style using text from the forgotten set, then employing the generated audio prompt to potentially address the issues mentioned in L234-236 about speaker confusion and privacy constraints.
3. There are inaccuracies in the bold results presented in Table 1; the WER-F and WER-R recorded for the TGU approach are higher than those of the Exact Unlearning and Fine Tuning methods.

**Questions:**

See the above weaknesses.

---

> ### Author Response · Authors · 2024-11-22
>
> We thank the reviewer for the time taken to thoroughly review our work and provide valuable feedback. We hope to take this opportunity to strengthen our work through your insights.
>
> ### **W3. There are inaccuracies in the bold results presented in Table 1; the WER-F and WER-R recorded for the TGU approach are higher than those of the Exact Unlearning and Fine Tuning methods.**
>
> We would like to address the concern labeled as 'Weakness 3' regarding the perceived inaccuracies in the bolded results in Table 1.
>
> In our initial submission, we highlighted the Word Error Rates (WER-F and WER-R) for the TGU approach, despite these figures being higher than those observed for the Exact Unlearning and Fine Tuning methods.
>
> We recognize that this decision might have conveyed an unintended message. Our rationale was focused on the comparative analysis within the subset of models that demonstrated a degree of unlearning (NG, KL, SGU, TGU). We apologize for any confusion this may have caused and are committed to correcting this in our revised manuscript to ensure clarity and accuracy in our comparative analysis.

---

> > ### Comment · Reviewer_r7jW · 2024-11-26
> >
> > Thank the authors for the reply. I'd like to regard Weakness 3 as a typo (the inappropriate bold results). Of course, the discussion about the performance drop and the trade-off of the proposed method should be added to the revised vision.
> > However, I'm looking forward to seeing the reply about Weaknesses 1 and 2, which are more important to the review of this paper.

---

> > > ### Author Response · Authors · 2024-11-28
> > >
> > > ### **W1. The paper currently lacks a discussion on these aspects, which are crucial for comprehensively evaluating the robustness and versatility of the proposed TGU framework.**
> > >
> > > We appreciate the reviewer's observations regarding the performance of TGU in robustness and versatility. We believe these experimental results will strengthen our paper and thank the reviewer for this discussion.
> > >
> > > ||SIM-R|SIM-F|
> > > |------|------|------|
> > > |Original|0.649|0.708|
> > > |TGU|0.631|0.169|
> > > |TGU*|0.593*|0.220*|
> > >
> > > *For this table, the last column is evaluated on specific remain and forget sets. The remain sets have high resemblance of forget speaker's audio (SIM greater than 0.4). The forget sets have low similarity to forget speaker's audio (SIM lower than 0.5). Please refer to Appendix C. of the paper for detailed information on ground truth SIM values of same speakers and different speakers.
> > >
> > >
> > > **The "fine-tune" process of TGU is based on given forgotten prompts, which naturally raises a question regarding how this framework performs with other prompts from the same forgotten speaker.**
> > >
> > > Our evaluation setup was designed such that the unlearned model does not encounter any prompts that were seen during training. When constructing the forget set, we excluded approximately 5 minutes of audio per speaker so it would not be used during training. The model was thus evaluated on other prompts not seen during training from the same forgotten speaker. However, we do acknowledge that a robustness and versatility experiment would benefit the paper.
> > >
> > > Inspired from reviewer's insights, we extended the experiment to exclusively measure performance on prompts from the same speaker with significantly different timbres. From the same speaker, although few, we noticed there were audio prompts with low similarity (under 0.5) when compared with other forget speaker's audio. We used these audio prompts to generate speech using TGU unlearned model to validate model's robustness. The generated speech returned a mean of 0.220 similarity with its audio prompt, proving that TGU was still able to unlearn speaker specific features and prevent a zero-shot generation. However, be noted that there were only 4 forget samples with low similarity to forget speaker's other audios.
> > >
> > >
> > > **Moreover, it is unclear how the framework handles prompts from other retained speakers who have similar timbres with a forgotten one.**
> > >
> > > To evaluate TGU’s robustness in handling remain speakers with high similarity to forgotten speakers, we identified remain samples that exhibited the highest speaker similarity (SIM) scores with any forgotten sample. We used these remain samples as prompts to generate speech with TGU and measured the similarity between the prompt and the generated output.
> > >
> > > We have provided a scatter plot result in the revised paper's Appendix G, Figure 6. The x-axis of the scatterplot represents the maximum SIM score between the remain audio with all forget samples, while the y-axis represents the similarity between remain audio (used as an audio prompt) with its TGU-generated speech. The results are as follows :
> > >
> > > - **Statistic : 0.1396**
> > > - **P-value : 0.0003**
> > >
> > > Apparently, there were no negative correlation between the x-axis and y-axis, indicating that even when remain speech depicted high similarity with forget speakers, the model was able to retain its speaker characteristics. Additionally, for remain samples that resembled forget speakers as much as over 0.4 SIM (x-axis), the average generated speech SIM is 0.593.
> > >
> > > The overall results indicate that TGU-generated outputs are independent of remain samples' similarity to forget speakers, and are generally robust in handling forget and remain identities. It is acknowledgeable that there is a performance drop in handling such identities of varying resemblance due to increased unlearning difficulty [1]. Yet, our TGU method was still able to selectively unlearn forget speaker specific characteristics to prevent generation, while retaining high performance for remain set.
> > >
> > > [1] Chongyu Fan, Jiancheng Liu, Alfred Hero, and Sijia Liu. Challenging forgets: Unveiling the worst-case forget sets in machine unlearning, 2024a. URL https://arxiv.org/abs/2403.07362.

---

> > > > ### Comment · Reviewer_r7jW · 2024-12-03
> > > >
> > > > I appreciate the authors' reply.  The response to weakness W1 reveals that the model's performance declines when dealing with prompts from other retained speakers who share similar timbres with a forgotten one. This insight suggests room for improvement in the model's robustness across varied speaker characteristics. Also, by combining the responses to W1 and W2, a question is raised: Could the principal results in the paper potentially be a result of overfitting the forget set? Hope the further exploration can be added to the revised version.

---

> > > > > ### Author Response · Authors · 2024-12-04
> > > > >
> > > > > We sincerely appreciate the reviewer highlighting areas for potential improvement and fostering thoughtful discussion. While we see the validity in the reviewer’s perspective, we would like to offer an alternative viewpoint.
> > > > >
> > > > > First, if the model were overfitting, one would expect it to only forget utterances explicitly present in the training dataset. For instance, an overfitted model would limit forgetting to the specific speech uttered by the target speaker within the training set. However, in our experiment, the model demonstrates **the ability to forget a range of diverse transcriptions spoken by the same speaker under different environments, suggesting a broader capacity beyond overfitting.** The unlearned model successfully recognizes the characteristics of the forget speaker apart from the remain even when SIM is low. Vice versa, the model distinguishes remain speakers even when they may sound similar in the terms of a Speaker Verification model.
> > > > >
> > > > > Second, we believe that the model’s sensitivity to the unique characteristics of the forget speaker reflects a more nuanced challenge rather than a straightforward case of overfitting - which typically implies a failure to generalize. Instead, we see this as an opportunity to engage in a deeper ethical discussion. For example, if the model forgets a particular speaker, should it also forget speakers with similar vocal characteristics? **Can the effort to protect one individual's privacy justifiably limit the technological opportunities available to others?** These questions extend beyond the technical realm and warrant careful ethical consideration.
> > > > >
> > > > > We greatly value the reviewer’s input in prompting these important discussions, which we believe contribute meaningfully to the ethical dimensions of this work.

---

> ### Author Response · Authors · 2024-11-28
>
> ### **W.2 The TGU mechanism resembles a distillation process, suggesting another potential baseline worth exploring.**
>
> We also thank the reviewer for suggesting another possible baseline for this research, and waiting for experimental results.
>
> **The approach could involve using the zero-shot TTS model to generate audio with the retained speaker style using text from the forgotten set, then employing the generated audio prompt to potentially address the issues mentioned in L234-236 about speaker confusion and privacy constraints.**
>
> ||WER-R|SIM-R|WER-F|SIM-F|
> |------|------|------|------|------|
> |Reviewer's Suggested Method| 2.3 | 0.571 | 2.3 | 0.197|
> |SGU | 2.6 | 0.523 | 2.5 | 0.194|
> |TGU | 2.5 | 0.631 | 2.4 | 0.169 |
>
> As the reviewer suggested, we utilized retained speaker style to generate audio prompt as new targets for this experiment. For each forget data $(x^{spk=f}, y)$ used to unlearn, we generated a target by randomly selecting audio prompt from the forget set. The generated target should then be an audio file uttering same text $y$ as the forget prompt with frame-wise alignment; $(x^{spk\not=f},y)$. While the results of the newly suggested baseline outperforms the naiive SGU, it is noticeable that TGU proposed in our paper still demonstrates effective unlearning while maintaining zero-shot performances on remain identities.
>
> We believe this is because RTU and SGU are more dependent on the remain set's speaker distribution to generate targets, while TGU uses randomly initialized noise. This strengthens our proposed TGU in addressing effective unlearning in ZS-TTS by implementation purely random behaviors - isolated from specific data distributions.
>
> We hope these experiments and discussions address your concerns and look forward to your answer.

---

### Meta-Review · Area_Chair_rdAo · 2024-12-17

**Metareview:**

The work presents TGU (Teacher-Guided Unlearning), an innovative mechanism designed to improve zero-shot text-to-speech (TTS) systems by addressing ethical and privacy concerns. Specifically, the authors propose a straightforward solution that involves fine-tuning the model on the original training set, using a newly defined target generated by the original teacher model without referencing the target speaker. As a result, the fine-tuned model maintains its performance on other speakers while producing random speaker identities for the voices that are intended to be removed.

Key claims are: (i) Unlike traditional unlearning methods, TGU incorporates randomness to prevent the consistent replication of forgotten speakers' voices, ensuring that unlearned identities remain untraceable; (ii) A novel evaluation metric, Speaker-Zero Retrain Forgetting (spk-ZRF), is introduced to assess the model's ability to effectively avoid reproducing forgotten voices.

The strenght of the work could be summarised as follows: (i) Origiality, since it is the first work in the field to define voice unlearning for zero-shot TTS (ZS-TT); (ii) Well written paper; (iii) Novel  metric spk-ZRF for evaluating speaker randomness in forget prompts.

Chief weakness of the work are: (i) Model performance declines when dealing with prompts from other retained speakers who share similar timbres with a forgotten one, (ii) The method requires the entire training data of the original model, 5 minutes of audio for the forget speaker, and more than a quarter of training iterations of the original model. (iii) Model performance degrades significantly as the number of “forgotten” speakers increase, (iv) Effectiveness across multiple TTS architectures, preferably open-source, should be evaluated.

 Assessing the proposed approach only on one single (old) architecture, limits a bit the broaness of the approach. Although,   the authors' justification on this is sound and respectable - new models were made available only in last 2024, some doubts about effectiveness of the approach remains. Furthermore, concerns on model performance degradation and (i and iii), and limited applicability (ii) remain even after the discussion phase. For example, when using a 1-minute instead of 5-minute clip, performance drop. Thefore,  the limitations in the training process is not a solved concern.

**Additional Comments On Reviewer Discussion:**

The discussion phase was quite lively and professional. It should be pointed out that the authors' rebuttal propted additinal comments from the reviewers. The authors replied to these comments.

Four independent reviewers evaluated the work, all acknowledging the novelty of the idea within the TTS domain. Reviewer zTeM was particularly positive in their initial review, although some of their concerns regarding weaknesses in the proposed approach were not fully addressed by the authors. Reviewers 98eY and ef5p maintained their reservations even after the discussion phase and recommended rejection. Similarly, Reviewer r7jW leaned towards rejection following the discussion.

 A Key concern is about the limited applicability of the proposed approach. Additionally, follow-up experiments spurred from the discussion phase  appear to confirm a performance drop in certain specific use-case scenarios.

---

### Decision · Program_Chairs · 2025-01-22

Reject